# Model-based lesion mapping of cognitive control using the Wisconsin Card Sorting Test

Jan Gläscher [1,2], Ralph Adolphs[2,3] & Daniel Tranel[4]

The role of the frontal lobes in cognition and behavior has long been enigmatic. Over the past decade, computational models have provided a powerful approach to understanding cognition and decision-making. Here, we used a model-based approach to analyze data from a classical task used to assess frontal lobe function, the Wisconsin Card Sorting Test. We applied computational modeling and voxel-based lesion-symptom mapping in 328 patients with focal lesions, to uncover cognitive processes and neural correlates of test scores. Our results reveal that lesions in the right prefrontal cortex are associated with elevated perseverative errors and reductions in the model parameter of sensitivity to punishment. These findings indicate that the capacity to flexibly switch between task sets requires the detection of contingency changes, which are enabled by a sensitivity to punishment that reduces perseverative errors. We demonstrate the power of model-based approaches in understanding patterns of deficits on classical neuropsychological tasks.

[1] Institute for Systems Neuroscience, University Medical Center Hamburg-Eppendorf, W34, Martinistr. 52, 20246 Hamburg, Germany. [2] Division of Humanities and Social Sciences, Caltech, M/C 228-77, 1200 E. California Blvd, Pasadena, CA 91125, USA. [3] Division of Biology and Biological Engineering, Caltech, M/C 228-77, 1200 E. California Blvd, Pasadena, CA 91125, USA. [4] Departments of Neurology and Psychological and Brain Sciences, University of Iowa, 200 Hawkins Dr, 2007 RCP, Iowa City, IA 52242, USA. Correspondence and requests for materials should be addressed to J.Gäs. (email: glaescher@uke.de)

The frontal lobes have long been thought to be critical for complex regulation of cognition, emotion, and behavior. The so-called "executive functions" that are compromised by damage to the prefrontal cortex (PFC) encompass aspects of cognitive control, planning, metacognition, and goal-directed decision making. These abilities depend on multiple functions, such as task switching, response inhibition, detection of performance errors and of response conflict, and working memory[1–3], typically measured by a wide array of neuropsychological tests (e.g., Stroop Test, Go-NoGo Task, Trail-Making Test, and Wisconsin Card Sorting Test (WCST)).

A goal in using of these tests in clinical applications is to provide both sensitivity and specificity to brain dysfunction, by serving as markers of particular cognitive processes that are engaged by the tasks. However, this goal is challenging because the tasks engage multiple cognitive processes. Nevertheless, the clinical relevance of such an approach could be considerable, given that frontal lobe dysfunction from traumatic brain injury (TBI) is a leading cause of disability in both the young and old, with an estimated 5.3 million people living with TBI-related disability in America[4], and given the prevalence of frontal lobe dysfunction in many degenerative neurological conditions (e.g., frontotemporal dementia)[5].

The WCST is one of the most frequently used measures of "executive functions"[6]. It was invented to formally probe the role of the prefrontal cortex in flexible behavior, notably the ability to shift between task sets[7]. Early neuroimaging studies of this task emphasized the involvement of dorsolateral prefrontal cortex (dlPFC) in set switching[8–10] and of the anterior cingulate cortex (ACC) in error detection[10] (see also refs. [11–13]). Nevertheless, it was clear early on that the task might depend on additional brain regions[14] and requires more than simply the ability to shift between sets. Indeed, it requires attention, working memory, abstraction, and decision making[10,15]. More recently, Wang et al.[16] emphasized the informational value of negative feedback in the WCST and demonstrated related activations in the (right) prefrontal and posterior cortices. Yet, none of these neuroimaging studies were able to identify the neural correlates of specific computations underlying performance of the WCST, since task performance engages multiple processes that are not isolated by standard cognitive subtraction approaches (see also Friston et al.[17] for an early critique).

In contrast, cognitive modeling is aimed at describing the constituent computations that underlie a task[18] and in combination with functional magnetic resonance imaging (fMRI; so-called "model-based fMRI"[19]) has been a powerful tool for pinning down the neural correlates of specific cognitive computations that contribute to task performances. In a recent example of this approach, Jiang et al.[20] were able to characterize a network comprised of the anterior insula, caudate nucleus, as well as ACC and dlPFC, as estimating the volatility of control demands, predicting the upcoming control demands and allocating attentional resources. Such specific characterizations of cognitive operations are beyond what can be achieved with the cognitive subtraction method, or with most standard scoring methods of the WCST and other neuropsychological tests. Nonetheless, it may be possible to re-analyze data from tasks such as the WCST using a model-based analysis, enabling the power of modern approaches to isolate its constituent processes.

The WCST[7,21] was initially conceived to measure and decompose complex decision making in the laboratory[22], and to parse the fuzzy construct of "executive functions"[23] (see Fig. 1 and Methods for a description). However, the standard decomposition of the WCST is rather qualitative and does not provide a link to parameters of a computational model. Instead, the WCST classically uses constructs such as "perseveration," "concept

formation," and "set maintenance," and reports summary measures such as perseverative errors, and the total number of categories obtained. These derived scores, while clinically useful, give little insight into which basic cognitive processes are responsible for test performance. For instance, perseverative errors may reflect specific deficits in processing feedback accurately, a lack of cognitive flexibility resulting in a failure to understand that task rules have changed, or a too narrow attentional focus on one specific dimension while failing to monitor others. Only very few studies attempt to decompose the processes engaged by the WCST into more fine-grained or computationally meaningful ones. Notably, Barcelo and Knight[24] proposed testing choice strategies through a further separation of non-perseverative errors into "efficient" and "random" errors, and Niv et al.[25] applied computational modeling to a version of the WCST with

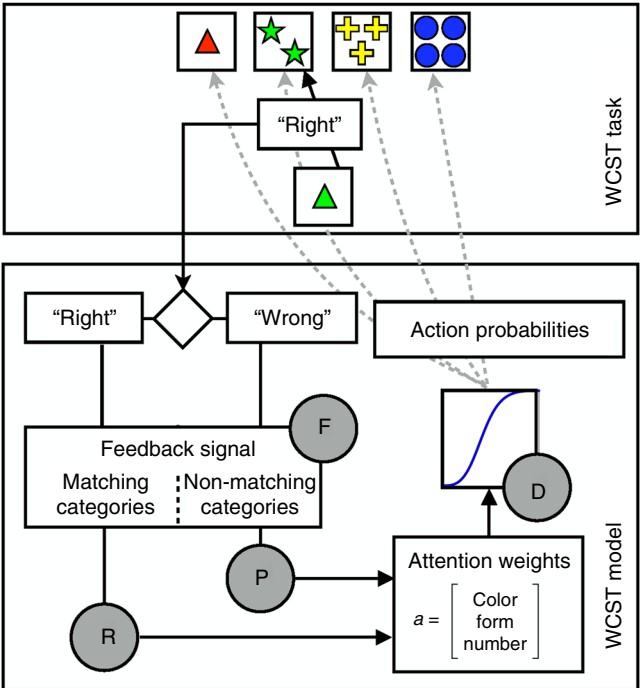

**Fig. 1** Schematic representation of the WCST and model. The WCST (top) requires participants to sort a sequence of cards into four piles, based on deterministic rules they need to learn from experience. In the example shown, a card with a green triangle could be matched to the piles based on color (green), form (triangle), or number (1). In the example, the prevailing current rule (determined by the examiner) is to sort by color, and the participants thus receive the feedback "right" if they put the card with the green triangle into the pile with the two green stars, and receive feedback "wrong" otherwise. Critically, the sorting rule is changed by the examiner after the participant sorts 10 trials correctly (unbeknownst to the participant and without any warning from the examiner). To perform accurately, the participant must learn what the new rule is (by trial-and-error sorting, based on the feedback they are given) and switch their strategy. The central components of the model of a participant's behavior on this task (bottom) are the attention weights, which represent the current belief about the relevant sorting rule. The attention weights are filtered with a sigmoid function, whose slope is controlled by the decision consistency parameter (D), to render action probabilities for each pile. Based on the actual choice of the participant and the feedback ("right" or "wrong"), a feedback signal is computed for either matching or non-matching categories; the signal is controlled by the attentional focusing parameter (F). The attention weights are then updated in proportion to the feedback signal weighted by the reward and punishment sensitivity parameters (R and P)

| Table 1 Number of participants and lesion volumes (ml) | | | | | | |
|---|---|---|---|---|---|---|
| **Cluster** | | **1** | **2a** | **2b** | **2c** | **3** |
| **No. of participants** | | **36** | **95** | **93** | **78** | **26** |
| **Region of interest** | **Hemi** | | | | | |
| Overall | | 29.71 | 32.88 | 40.24 | 35.23 | 57.26 |
| Prefrontal cortex | L | 11.14 | 9.93 | 16.21 | 11.40 | 19.61 |
| | R | 12.38 | 14.39 | 22.30 | 16.37 | 17.79 |
| Posterior PFC | L | 1.13 | 2.09 | 3.81 | 4.79 | 2.67 |
| | R | 5.04 | 4.28 | 5.54 | 4.82 | 8.22 |
| Parietal cortex | L | 8.71 | 8.05 | 6.27 | 6.66 | 8.96 |
| | R | 4.61 | 7.53 | 10.82 | 8.11 | 44.52 |
| Ant. temporal lobe | L | 16.66 | 11.02 | 13.45 | 11.35 | 16.60 |
| | R | 13.68 | 14.80 | 13.01 | 7.53 | 16.31 |

Overlap (ml) of each lesion with the listed regions of interest averaged across participants in each cluster. ROI definitions: prefrontal cortex comprises frontal pole, OFC, vmPFC, dlPFC, and anterior ACC; posterior frontal cortex comprises insula and precentral gyrus; parietal cortex comprises postcentral gyrus and supramarginal gyrus, and superior parietal lobule; anterior temporal cortex includes temporal pole, anterior middle and superior temporal gyrus, and anterior parahippocampal gyrus

probabilistic feedback (neither of the approaches of those two studies were applicable to our data).

The limitations in terms of experimental design and analysis have been compounded by statistical and conceptual problems as well[26–33]. Many earlier studies (both lesion and fMRI) were seriously underpowered in terms of sample size (typically $N < 50$, and often much smaller). Some studies even failed to demonstrate a reliable association with prefrontal damage altogether, sometimes despite reasonably large sample sizes[34]. An even more fundamental problem pertains to the correlative nature of neuroimaging studies, which cannot differentiate between brain regions that are necessary for task performance and those which are not[35]. Given a large enough sample size of patients with focal, chronic brain damage, methods such as voxel-based lesion-symptom mapping (VLSM[36]) can identify neuroanatomical regions necessary for task performance, based on stable, long-lasting behavioral impairments evident even after cortical reorganization following brain damage has been completed[37].

The overarching aim of the present study was to provide a decomposition of WCST performance into processes that correspond to the parameters of a computational model, and to use the variability across these parameters caused by frontal lobe damage to fractionate dysfunction in ways that could ultimately help with clinical diagnosis and management. Our approach, model-based lesion mapping, seeks to fractionate the cognitive component processes engaged by the WCST onto neuroanatomical sectors of the PFC[38], using a sample size of 328 patients with focal lesions to the PFC and more posterior brain regions. This sample is considerably larger than any prior lesion mapping study of the WCST[27,28,31,39,40]. We apply a computational model based on prior work by Bishara et al.[41], and we use voxel-based lesion-symptom mapping[19,36,42] in order to map parameter estimates from the model to specific brain regions.

## Results

**Lesion distribution**. We tested 328 patients who completed the WCST and who had a single, focal, chronic brain lesion. Lesions were mostly centered in prefrontal, frontopolar, and posterior frontal regions, but more variably also included anterior parietal cortex, anterior temporal lobe, as well as more posterior brain regions (Table 1). The data in Table 1 are separated into five

different clusters of participants that we identified in a later two-stage cluster analysis of the model parameters from our model (see below). The right hemisphere was sampled more densely than the left (Supplementary Figure 1), which may have been due in part to the exclusion of some patients who were too aphasic to yield a valid performance (see below). All patients completed background measures of intelligence and other neuropsychological tests (listed in Supplementary Tables 1 and 2), as well as the WCST.

**Computational model**. We then used the computational model of Bishara et al.[41], which explains the observed WCST performances in terms of four underlying cognitive processes. The model focuses on how attention is deployed to specific dimensions of the WCST that determine the correct rule by which to sort (color, form, number), and how attention is changed in the face of feedback from the examiner. Figure 1 (bottom) shows a simplified schematic of the model. Supplementary Figure 2 shows the full graphical model used for Bayesian estimation. Note that each participant was fitted individually, so that each participant's set of parameters was fully independent of those of other participants, thus ensuring validity in the subsequent leave-one-out cross-validation analysis.

**Parameter recovery study**. We first validated the computational model by quantifying its ability to recover true parameter values from simulated data (see Supplementary Note 3). We systematically varied the values of all 4 model parameters, simulated 60 virtual participants for each parameter combination, and fitted each of these simulated data sets individually using our Bayesian parameter estimation (Supplementary Figure 2). In general, the model was very accurate in recovering the true parameter value (Supplementary Figure 3), although it tended to underestimate the true values for medium and high reward and punishment sensitivity parameters. This simulation study confirmed that the Bishara et al.[41] model we used is able to recover the true parameter values, validating its application to our empirical patient data.

**Relationship between model parameters and WCST scores**. We next fitted our WCST data from 328 lesion patients with this model using Bayesian estimation (Supplementary Figure 2) and correlated the model parameters with five selected WCST scores. Figure 2a displays boxplots of WCST scores with reference to published norms[21], whereas Fig. 2b shows a box plot of the fitted model parameters with reference to the healthy comparison group reported by Bishara et al.[41] (orange lines in the figure). Figure 2c shows the correlations among and between all these data. The highest correlation occurred between the punishment sensitivity (P) and decision consistency (D) parameters of the model. These were also the two model parameters that correlated substantially with perseverative (PSV) and non-perseverative errors (NPSV) and with number of categories achieved (NCAT) from the original WCST scores. These correlations suggest that the P and D parameters of the model may be most informative for explaining deficits on the WCST scores, in particular for perseverative errors.

In order to validate that the model accurately predicted our observed scores (posterior predictive check), we simulated 50 data sets for each of our 328 patients using the individual fitted parameters (maximum a posterior estimate) and computed the average WCST scores for these virtual participants. Comparing these predicted scores with the observed scores (Fig. 3) confirmed that the model was very accurate in reproducing the data.

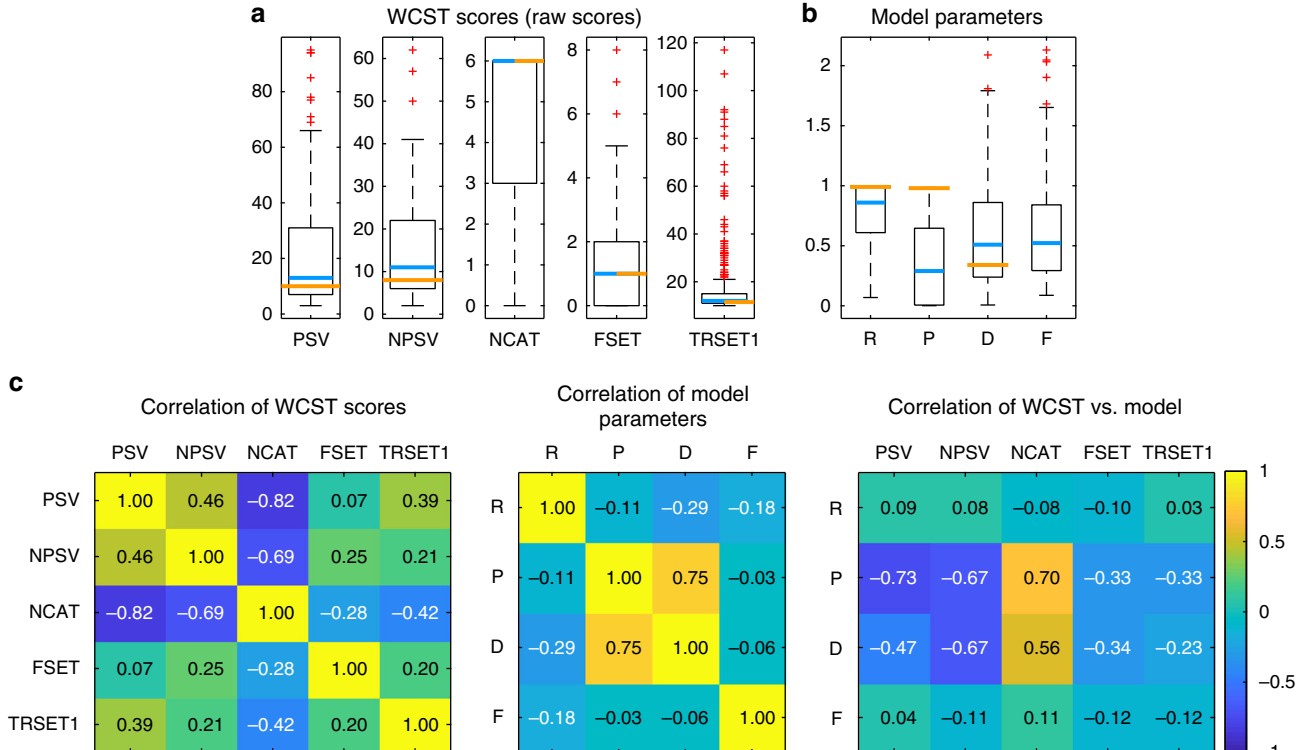

**Fig. 2** WCST scores and model parameters from our participant sample. Boxplots of 5 WCST scores (**a**) and 4 model parameters (**b**). PSV perseverative errors, NPSV non-perseverative errors, NCAT number of categories achieved, FSET failure to maintain set, TRSET1 trials to complete first set, R reward sensitivity, P punishment sensitivity, D decision consistency, F attentional focusing. Upper and lower boundaries of the boxes indicated the 75th and 25th percentile, whiskers extend to the most extreme data points not considered an outlier, whereas outliers are plotted as red crosses. Blue lines indicate the median of our sample and orange lines indicate reference points for comparison (in **a** orange lines represent the median from demographically corrected norms[21] and in **b** orange lines represent the median from healthy comparison participants reported by Bishara et al.[41]). The F parameter was fixed in that study and hence is not reported. **c** Correlation among and between WCST scores and model parameters

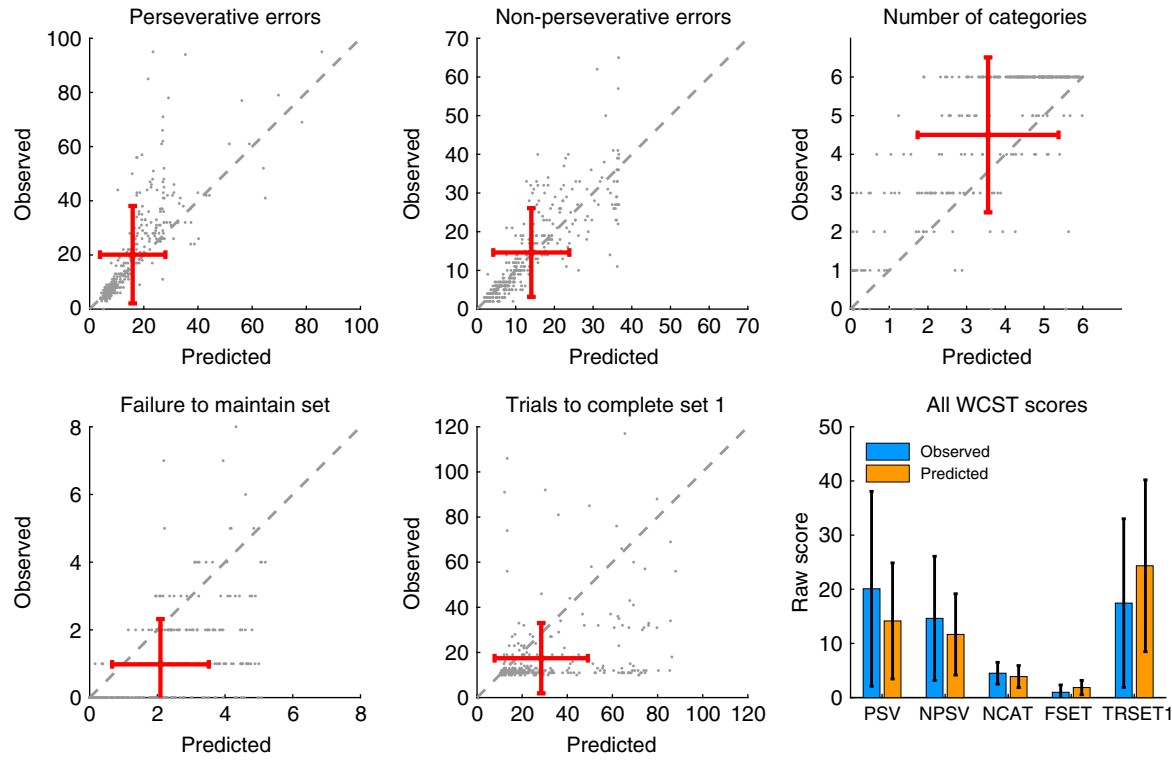

**Fig. 3** Comparison of observed vs. model-predicted WCST scores (posterior predictive check). Red error bars indicate the mean of the observed and predicted WCST scores, which are also shown more conventionally in the bar plot (error bars are ±1 s.d.)

**Table 2 Model comparison**

| Model variant | Description | Num. param. | ΔDIC |
|---|---|---|---|
| RPDF | Full model (as described in text) | 4 | 61,257.59 |
| RRDF | Same (estimable) parameter for reward and punishment sensitivity | 3 | 58,698.38 |
| RP1F | Decision consistency fixed at 1 (same and moderate decision noise across participants) | 3 | 60,355.39 |
| RPD0 | Attentional focusing fixed at 0 (equal weight to the update of attention weights) | 3 | 59,845.54 |

The deviance information criterion (DIC) for several (full and degenerate) versions of the Bishara model is shown as a difference to a null model of random test performance (ΔDIC). Higher ΔDIC indicates better model performance

**Effects of model parameters on perseverative errors**. To further explore the relationship between model parameters and perseverative errors, we conducted another simulation grid search study, in which we systematically varied all four model parameters across their entire or reasonable range, simulated synthetic data for each parameter combination, and calculated the standard WCST scores (see Supplementary Note 4). Because of their theoretical and clinical importance, we focus on perseverative errors (PSV) and present their dependency on different combinations of model parameters in Supplementary Figure 4. These graphs reveal that PSV are minimized when reward sensitivity (R) is low, punishment sensitivity (P) is high, decision consistency (D) is high, and attentional focusing (F) is low. Thus, in line with the correlations between model parameters and WCST scores (Fig. 2c), this simulation study also attributes an important role for punishment sensitivity and decision consistency in the generation of PSV in brain-damaged patients.

**Model comparison**. We tested the full Bishara model against three degenerate versions of the model that fixed one of the four parameters to a constant value, to test whether each of the freely estimable parameters of the model are necessary (Table 2 and Methods). These values demonstrate that the full Bishara model resulted in a superior model fit compared with the three other versions.

**Voxel-based lesion-symptom mapping of model parameters**. We then submitted all model parameters and WCST scores to a univariate VLSM analysis[36,42] (see Methods, Supplementary Note 5 and Supplementary Figures 5–7 for details). Of the four model parameters, only punishment sensitivity (P) resulted in a significant lesion effect, which was located primarily in the right PFC reaching from dorsolateral PFC to the frontal pole and mostly focused in the underlying white matter (Fig. 4 and Table 3). No significant effects were found for the R, D, and F parameters. Of the 5 WCST scores only PSV (Fig. 4 and Table 3) and trials to complete first set (TRSET1, Supplementary Figure 5) exhibited significant lesion effects, localized in right PFC. We found a large degree of overlap between the statistical maps for PSV and P (Fig. 4, bottom row). The co-localization of PSV and P in terms of their lesion effects reinforces the importance of punishment insensitivity in the generation of perseverative errors on the WCST.

Neuropsychological task performance is often affected not only by impairments caused by specific neuroanatomical lesions, but also by more general or co-occurring factors such as total lesion volume or impairments on other neuropsychological tasks. In a control analysis, in which we regressed out the variance of lesion volume, demographic factors, and neuropsychological measures listed in Supplementary Tables 1 and 2, we again found significant effects for punishment sensitivity and perseverative errors (at a lower threshold of 5% false discovery rate (FDR)) in the right PFC (Supplementary Figure 6). The general pattern of overlapping lesion correlates for these two measures (P and PSV) in the right PFC remained intact.

Another important consideration, in any lesion mapping study, is the inhomogeneous density of lesions of the dataset, resulting in unequal statistical power to detect significant relationships in different parts of the brain. Given that our lesion density was in fact highest in the right PFC (Supplementary Figure 1), the very region where we also report effects, we thus carried out another control analysis in which we equated lesion density. This was done by choosing a subset of $N = 239$ patients whose composite lesion density map was homogeneous in the left and right PFC. When we carried out the same analyses on this subset of patients, we obtained very similar results (Supplementary Figure 7): perseverative errors and punishment sensitivity also overlapped to a large degree in the right dorsolateral prefrontal cortex and underlying white matter.

Finally, to overcome limitations in statistical power in posterior brain regions we further explored the model parameters and WCST scores of 13 patients with left posterior lesions and compared them to 27 patients with right posterior lesions (an area for which we did have sufficient statistical power). This analysis revealed that for patients with left or right posterior lesions, there were almost identical profiles of model parameters and WCST scores (Supplementary Figure 8). This outcome suggests that damage to left posterior cortex can also result in specific impairments in the WCST, and that right posterior involvement may not be a unique feature. However, we emphasize that this conclusion remains speculative, as we did not have sufficient power in left posterior cortex to assess statistical reliability.

**Cluster analysis of model parameters**. We then sought to identify subgroups in our sample with distinct profiles of model parameters. A $k = 3$ means clustering provided the best initial fit to the data (Supplementary Note 6, Supplementary Table 3, and Supplementary Figure 9). Plots of the average model parameters and WCST scores for the participants in each cluster reveal anatomical locations associated with the task-derived clusters (Fig. 5). Judging from the performance profiles of these clusters on the WCST and other neuropsychological background tests (Supplementary Table 2), this cluster analysis primarily identified a small group of very good performers (cluster 1, $n = 36$) and a group with poor performance on many neuropsychological tests (including the WCST) (cluster 3, $n = 26$), while lumping all other participants into a remaining large group (cluster 2, $n = 266$), without any differentiation within different PFC regions. Consistent with our exploration of the model parameter space (Supplementary Figure 4), participants in cluster 1 also exhibit the greatest punishment sensitivity (P) and decision consistency (D) parameters. These clusters exhibited marked differences in their respective density maps: whereas lesions in cluster 1 were focused on the temporal poles, lesions of the other two clusters were primarily found in the right PFC and parietal cortex.

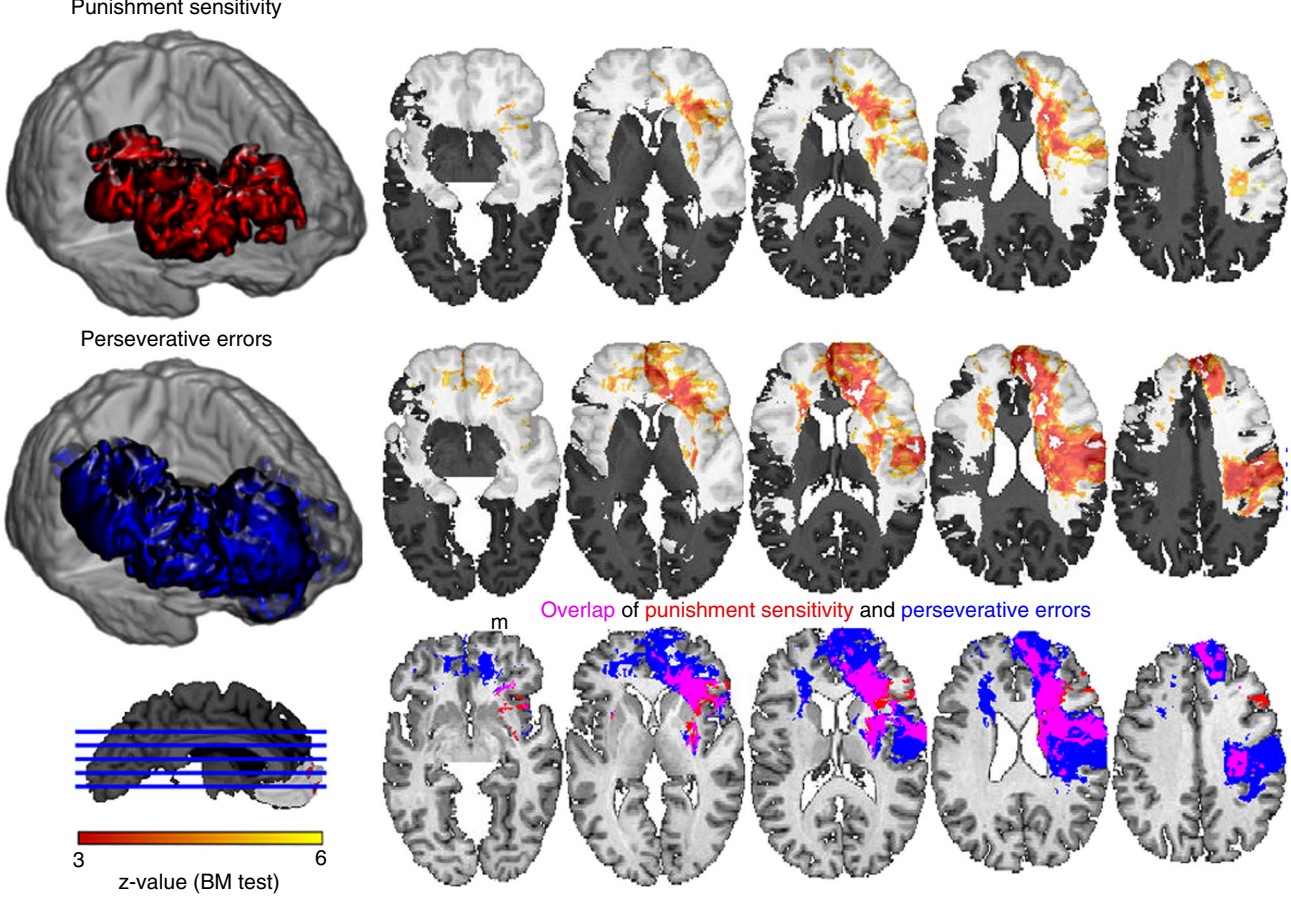

**Fig. 4** Univariate lesion mapping of WCST scores and model parameters. Punishment sensitivity (P) and perseverative errors (PSV) showed a significant lesion effect in right PFC encompassing dlPFC and frontal pole and stretching to posterior frontal and parietal cortex. Individual results for both variables are shown on transverse slices (the right hemisphere is on the right in these slices). Shaded areas indicate regions of insufficient statistical power given our 1% FDR threshold. The three-dimensional (3D) shapes of the lesion effects are shown in the colors used in the overlap slices. The overlap in the bottom panels corresponds to a conjunction analysis. Note that this is not a statistical interaction term; the three colors simply denote overlap or non-overlap, not statistical independence. Scale: z-score obtained from the Brunner–Munzel test implemented in NPM/MRIcron

**Table 3 MNI coordinates of peak voxels for different VLSM analyses**

| Region | Hemi | X | Y | Z | Z-value |
|---|---|---|---|---|---|
| **Punishment sensitivity** | | | | | |
| Anterior corona radiata | R | 25 | 21 | 23 | 4.88 |
| Superior corona radiata | R | 26 | −4 | 22 | 6.00 |
| Superior fronto-occipital fasciculus | R | 21 | 8 | 22 | 5.33 |
| Anterior corona radiata | R | 22 | 29 | 7 | 4.94 |
| **Perseverative Errors** | | | | | |
| Anterior corona radiata | R | 18 | 36 | 16 | 5.22 |
| Superior frontal gyrus | R | 15 | 54 | 18 | 5.16 |
| Superior frontal gyrus | R | 15 | 50 | 24 | 4.87 |
| Medial superior frontal gyrus | R | 14 | 50 | 31 | 5.10 |
| Anterior corona radiata | R | 21 | 21 | 18 | 4.96 |
| Superior longitudinal fasciculus | R | 35 | −2 | 23 | 5.36 |
| Postcentral gyrus | R | 62 | −8 | 25 | 5.11 |
| Lingual gyrus | R | 62 | −4 | 14 | 5.09 |
| Anterior corona radiata | L | −23 | 24 | 12 | 4.85 |

To gain further anatomical differentiation of the large cluster 2, we carried out a second $k$-means cluster analysis ($k = 3$, see Supplementary Note 6, Supplementary Table 4, and Supplementary Figure 10) exclusively on the participants from cluster 2 (Fig. 6). These newly derived clusters had varying extents of prefrontal lesions (2b > 2c > 2a) (Table 1 and Supplementary Figure 12), which were matched by a decreasing performance on the WCST in these groups (2b < 2c < 2a) when considering PSV, NSPV, and NCAT. This suggests that the extent of damage to anterior PFC is associated with the degree of impairment on the WCST (Table 1). The average lesion overlap of cluster 2b in the right and left PFC and in right parietal cortex was larger than for clusters 2c and 2a. Taken together, these findings confirm the critical role of the right PFC and right posterior cortices in implementing the cognitive processes required to perform well on the WCST, and they also suggest that the extent and location of damage within right PFC and posterior cortices may produce somewhat different profiles of impairment and graded severities of impairment. We validated the cluster solution of this two-step procedure in a cross-validation approach (Supplementary Note 7, Supplementary Figure 11, and Supplementary Table 5) and found high agreement between model parameters and WCST scores in the cross-validation samples and the original solution suggesting that the cluster solutions presented in Figs. 5 and 6 are replicable and valid.

## Discussion

We applied a computational model to data from the Wisconsin Card Sorting Test in what is, to our knowledge, the largest lesion mapping sample to date. Both the computational model and the

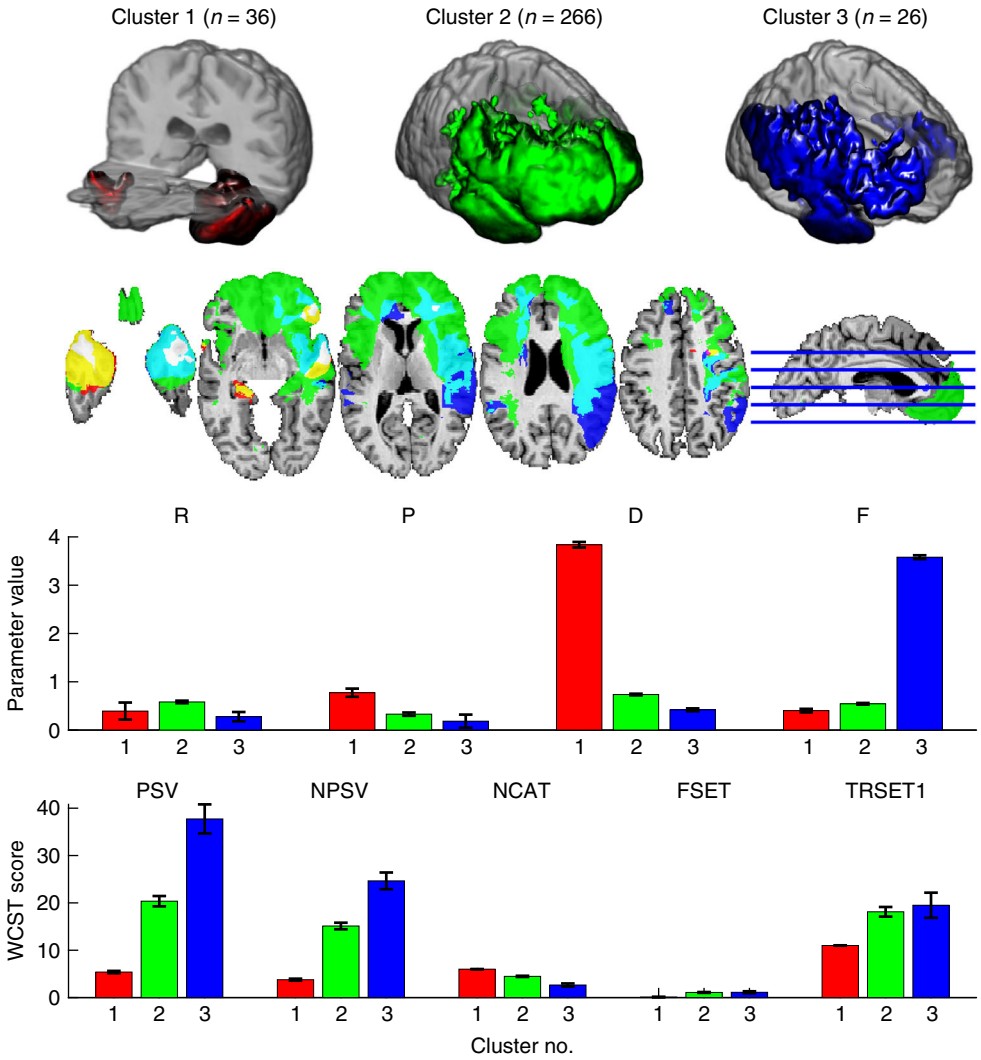

**Fig. 5** Cluster analysis of model parameters. Lesion density maps (thresholded at >8 lesions) for each cluster. The focus of each density map is in the right vlPFC with further involvement of right dlPFC and left frontal pole in clusters 2 and 3. Distinct profiles of model parameters and WCST scores corresponding to the three clusters (model parameters: mean of individual maximum a posterior (MAP) estimate ± s.e.m., WCST scores: mean ± s.e.m.)

VLSM approach provided a detailed fractionation of processes engaged by the WCST and a visualization of their essential neuroanatomical correlates in the human brain. Simulation studies that we performed provide further validation of the cognitive model employed in this study.

These findings showcase the approach (model-based lesion mapping), uncover a key component process most responsible for impaired task performance (punishment sensitivity), and identify brain regions wherein damage is most strongly associated with impairment (right prefrontal and frontoparietal cortices and underlying white matter). Our results also underscore the considerable heterogeneity across lesion–behavior associations, and the fact that a broad anatomical range of lesions including right (and to a certain degree left) posterior cortices can be associated with impaired task performance.

The two-step cluster analyses identified several subgroups with specific patterns of model parameters and different degrees of impairments. One intriguing potential future direction for these data would be to generate something like a cognitive "fingerprint" that could be diagnostic of lesion anatomy. In principle, one could use the similarity in the profiles of model parameters to make predictions about lesion location in specific brain regions. Such a pattern of impairment across processes (like the ones

shown in Figs. 5 and 6) could accurately classify subgroups of patients with lesions. A limitation that remains in such long-range objectives is statistical power, due to restricted sample size, and this could in principle be overcome through data sharing across laboratories.[43]

Neuroimaging work has shown that the WCST activates a network of brain structures that include not only sectors of prefrontal cortex, but also posterior cortices[9,10,14], whereas lesion evidence by and large has continued to suggest a disproportionate role for the PFC, in particular the historically dominant role of dorsolateral PFC[30], despite evidence for additional posterior involvement[27,39]. Thus, while the WCST is accepted as a sensitive probe of frontal lobe dysfunction, there is also consensus that it is not a very specific one[15,44]. A few studies have even reported that PFC lesions do not impair WCST performance at all, or that they impair it no more than lesions elsewhere[31,38,45].

At least two shortcomings of all prior studies are worth noting here, both of which we addressed in our present study. First, none of these prior studies used voxel-based lesion mapping, but rather mapped lesions classically and often with quite small sample sizes; this would considerably limit anatomical specificity and statistical power. Second, no prior lesion study used a model-based approach to the WCST. Instead, prior lesion studies were

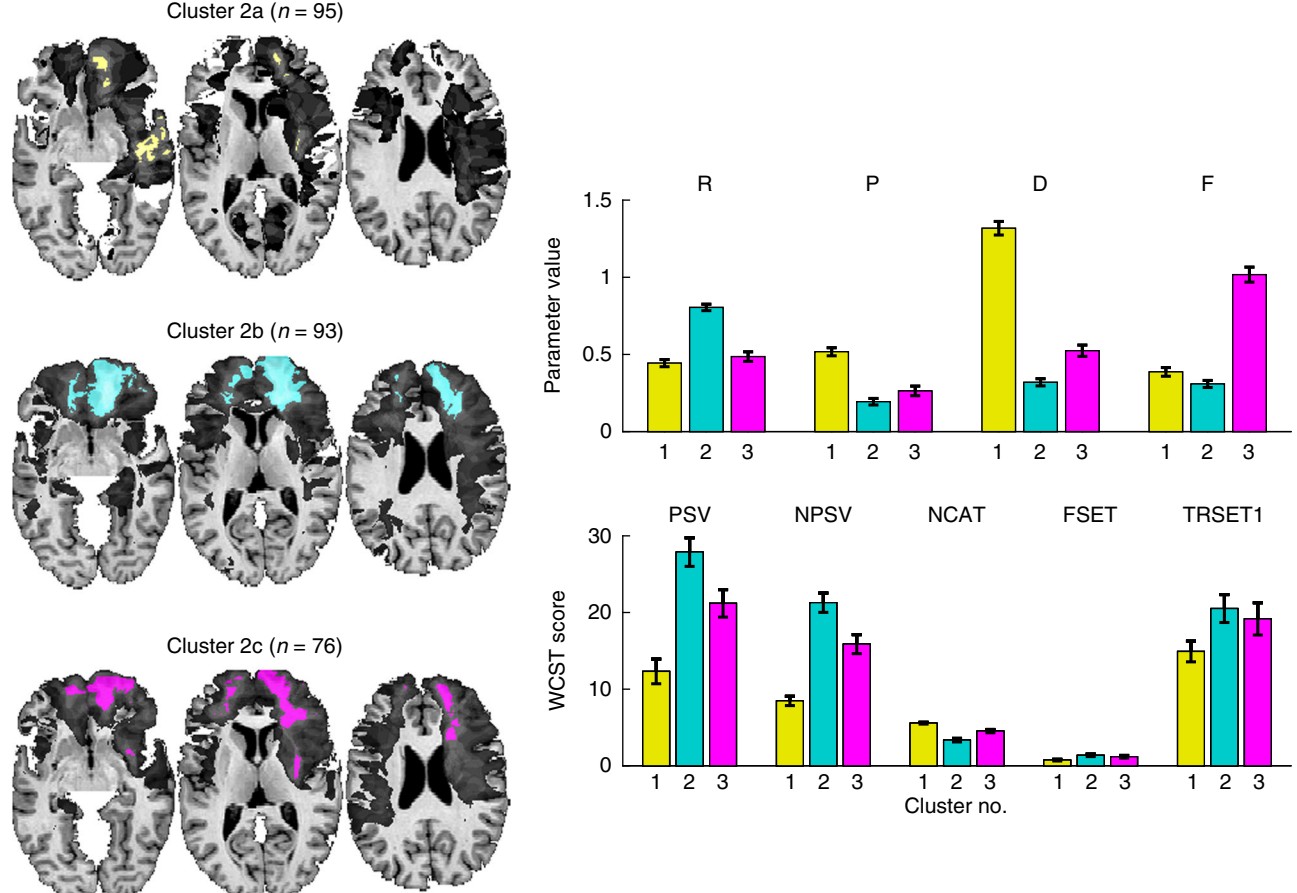

**Fig. 6** Cluster analysis of participants derived from cluster 2 of the first analysis (Fig. 5). This second analysis identified three further subgroups of participants who differed in their model parameters and WCST scores (mean individual MAP estimate ± s.e.m.)

based on conventional scores from the WCST, leaving open the important question of what kinds of measures to map onto the brain in the first place.

There is general agreement that the frontal lobes are required for cognitive control processes[1], and that this level of control is near (or at) the top of a set of hierarchically ordered processes[46]. However, making more fine-grained distinctions among processes, and among specific sectors of the PFC that are putatively associated with such processes, has been more difficult. In terms of lateralization, a key role of right PFC in the WCST has been evident since the earliest neuroimaging studies[8,10,47], whereas results from lesion studies have been more mixed, with some even pointing more to the left than the right PFC[28]. Also, right inferior frontal gyrus (IFG) involvement has been often interpreted as a major neural correlate of inhibitory control[48,49], although others associate it more with attentional control to salient stimuli[50].

While our present findings are consistent with the predominant idea of right IFG involvement in key executive function components, they also extend this idea in two important respects. First, the critical brain regions: (a) extended posteriorly, encompassing parts of anterior parietal cortex, and (b) substantially involved white matter—indeed, to such an extent that underlying white matter disconnection within the right frontal lobe may play a greater role than damage to cortex per se (see Figs. 4–6). This is consistent with previous studies highlighting the role of white matter connections in higher-order cognition, such as other decision-making tasks[51,52], general intelligence[53], processing speed[42], and emotional processing[54]. White matter involvement is also highly relevant from a clinical perspective, as one of the

most common causes of frontal lobe damage, traumatic brain injury, often involves diffuse axonal injury[55].

A second important extension of our study is the finding of punishment insensitivity as one of the core components isolated by damage to right IFG (Figs. 4–6). Sensitivity to negative feedback is an essential ingredient of successful performance on the WCST: it helps to detect expectancy violations and therefore reduces perseverative errors that result from failing to flexibly switch between sorting dimensions. This task requirement resembles numerous real-world situations in which response contingencies change suddenly and often rather capriciously. It can be proposed, for example, that computationally, punishment sensitivity amplifies the negative prediction errors resulting from negative feedback, facilitating a shift in attention weights and—as a consequence—a behavioral switch to another sorting dimension. This is underlined by our correlational analyses and simulation studies, which demonstrated strong negative correlation between perseverative errors and punishment sensitivity (Fig. 2c) and a reduction in perseverative errors under high punishment sensitivity (Supplementary Figure 4).

There are important limitations to our study. The anatomical findings are limited by our sample size and the spatial distribution of lesions within our sample, and the conclusions about processes are limited by the specific process model[41] that we used. A larger and/or differently distributed set of lesions could yield somewhat different, and possibly more finely differentiated, anatomical findings; a different computational model could yield alternative cognitive processes.

Our patient sample included a strong representation of frontal and right-sided lesions, and we had less representation of

posterior lesions on both sides of the brain. It will of course be important to extend the present findings to larger samples, and to samples that include lesions from varied parts of the brain, including in posterior regions. In principle, this could lead to findings at anatomical locations we did not have the power to detect in our present study. However, we would not expect radically different findings—for example, as shown in our control analysis in Supplementary Figure 7, equating the lesion distribution in fact did not grossly change our findings. While our findings are limited to the regions of the brain sampled, they are thus likely to hold up as robust. No less important from a clinical perspective will be an examination of lesion etiology. While our study only used participants with focal brain lesions, and thus excluded those with diffuse brain injury, the latter of course constitute by far the largest proportion of individuals with brain damage (e.g., from traumatic injuries or degenerative conditions), and could well be examined using our approach.

Another important consideration is the nature of the computational model. As our results demonstrate, we were careful to ensure that our model is well behaved, that it can capture a large range of performance, and that it can regenerate standard WCST scores well. But we did not test any alternative model. It is quite conceivable that models predicated on other processes, or models with more complexity, could perform as well or better. The reason that we chose the model of Bishara et al.[41] in the first place was that it is prima facie very plausible, and parsimonious. While there are certainly more complex models, it is difficult to imagine ones that are much simpler and yet still capture the basic structure of the WCST—indeed, reducing the model to three (instead of four) free parameters greatly reduced model fit in our study (cf. Table 2). The model should thus be thought of as the best starting point, with possible elaborations once additional evidence supporting such elaborations would emerge.

While previous studies have mapped out scores obtained from a factor analysis of neuropsychological tests of spatial neglect[56], the present study combines dedicated computational modeling with lesion mapping into an innovative analysis framework, an approach used only rarely before[57]. Whereas model-based fMRI[19] is primarily focused on identifying the neural correlates of core computational signals derived from the internal variables of a model as they unfold over time, model-based lesion mapping is aimed at identifying neural signatures of variance in model parameters across participants (e.g., punishment sensitivity), whose modulatory influences shape these core computations (e.g., attention weights). Our findings could be complemented by applying the Bishara et al.[41] model to fMRI data from the WCST, possibly combining fMRI-based analyses with lesion-based analyses. Connectivity analyses between these areas and the ones reported here, which ostensibly represent the modulatory influences on these core computations, could further highlight the neurobiological systems supporting performance on the WCST. An intriguing possibility would be to combine fMRI and lesion analyses in the same participants, e.g., by having patients with lesions perform a WCST-type task in the scanner. Such analyses could yield important insights not only into degeneracy in how cognitive processes map onto brain regions[58], but also reveal compensation and reorganization in neurological patients.

Another important next step in model-based lesion mapping is to move from univariate (e.g., VLSM) to multivariate analysis techniques[59,60], as the latter exhibit superior sensitivity and are therefore able to detect more subtle lesion-deficit associations, such as the hemispherically comparable impairments we observed in bilateral posterior brain regions (Supplementary Figure 8). In particular, approaches based on canonical correlation[61] or game-theoretic measures[62,63] could relate a multivariate pattern of lesion damage to a multivariate pattern of deficits on task

performance, and even estimate the individual contribution of each damaged region to the observed behavioral deficits

Finally, our findings could be used to inform and revise the application of the WCST. For instance, computational modeling could be combined with classical scoring of the WCST to yield a new scoring algorithm—one not intended to replace, but rather augment the current scoring scheme with further information about the cognitive processes involved in task performance. Also, our WCST dataset did not contain information about reaction time (such data are not routinely collected in conventional administrations of the WCST), and RT data could potentially be an important addition in future models of this task. Moreover, if model-based scoring algorithms were applied to different clinical populations, this could produce a "cognitive fingerprint" for different types and patterns of cognitive dysfunction, as suggested in the cluster analysis we present in Figs. 5 and 6. This effort might not be restricted to individuals with neurological conditions, but could potentially be applied to other populations (e.g., individuals with psychiatric disorders, developmental disorders). In that sense, our approach using model-based lesion mapping parallels efforts in the emerging field of Computational Psychiatry[64]. This could, together with other neuropsychological data, help to generate a scientific ontology of cognitive impairments. While incomplete in many respects, we believe the present study shows promise towards that goal.

## Methods

**Participants**. We analyzed a final dataset from 328 neurological patients (171 male and 157 female participants, mean age at testing 55.8 years, range 20–89 years; see Supplementary Table 1) who were evaluated under the auspices of their enrollment in the Iowa Neurological Patient Registry. This sample includes 186 patients from a previous analysis of the WCST[40]. All patients had been extensively characterized in terms of their neuropsychological and neuroanatomical status. All patients had a single, focal, chronic lesion in the brain. Patients with progressive diseases or psychiatric illnesses and those with diffuse lesions were not included (those conditions are exclusion criteria for being enrolled in the Patient Registry). We also excluded patients who had aphasia of such severity as to interfere with comprehension of the WCST instructions and preclude valid WCST performance. Specifically, we excluded 3 such patients, based on the dual criteria of having scores <35 on the Token Test and scores <15 on the Aural Comprehension Test from the Multilingual Aphasia Exam. All participants gave written informed consent at the time of their enrollment in the Iowa Neurological Patient Registry and the study was approved by the University of Iowa Institutional Review Board. Supplementary Table 1 provides demographic information about our sample.

**Wisconsin Card Sorting Test**. All patients completed the standard 128-item (2 identical decks of 64 cards each), hand administered version of the WCST[6,21]. The task requires the participant to sort cards with symbols that can be characterized in term of 3 sorting dimensions (Color, Form, Number) into 4 piles. The experimenter provides explicit feedback about the correctness of the participant's choice, by saying "right" or "wrong" (note that the 1993 manual[21] allows either "correct" v. "incorrect" or "right" v. "wrong," but we use "right" and "wrong" with our patients to avoid any ambiguities of patients hearing accurately the small difference between "correct" v. "incorrect"). This feedback, which provides a strong social reward or punishment, can be used by participants to update their choice strategy. After 10 consecutive correct sorts, the sorting dimension changes, unbeknownst to the examinee (if the examinee completes 6 category sorts of 10 each correctly, the test is discontinued). Using trial and error, and learning from the feedback provided by the experimenter, the participant can infer the correct sorting dimension. Based on their ubiquitous usage in the field, we selected the following WCST scores for validating and comparing the results of the computational modeling: (1) perseverative errors (PSV), (2) non-perseverative errors (NPSV), (3) number of categories achieved (NCAT), (4) failure to maintain set (FSET), and (5) trials to complete set 1 (TRSET1) (for a description of these indices, see Supplementary Note 1).

**Computational model**. We chose the model presented by Bishara et al.[41] as the computational framework for this study. This model has 4 free parameters: 1. reward sensitivity (R; the sensitivity to the feedback "right"), 2. punishment sensitivity (P; the sensitivity to the feedback "wrong"), 3. decision consistency (D; how much the choice is influenced by the attention weight), and 4. attentional focusing (F; the degree to which the update is focused on only the dimension with the largest attention weight). The model computes the probability to choose the selected pile as a function of "attention weights" toward each sorting dimension and how well

the current card is matching with the exemplars of the selected pile. These attention weights are updated according to a feedback signal that also depends on the match between the current card and the selected pile. Concretely, action selection (choice of pile) is done using the power form to computed action probabilities

$$\mathbf{P} = \frac{\mathbf{m}_t' \mathbf{a}_t^d}{\sum \mathbf{a}_t^d}, \tag{1}$$

where $\mathbf{m}_t'$ is a transposed $3 \times 1$ vector encoding matches ($=1$) or non-matches ($=0$) between the current card and the selected pile; $\mathbf{a}_t$ is a $3 \times 1$ vector of attention weights (range between 0 and 1), and $d$ is a free decision consistency parameter.

The feedback signal computes the amount by which the attention weights should be updated given the outcome of the current trial.

$$\mathbf{s}_t | \text{right} = \frac{\mathbf{m}_t \mathbf{a}_t^f}{\sum \mathbf{m}_t \mathbf{a}_t^f} \quad \mathbf{s}_t | \text{wrong} = \frac{(1 - \mathbf{m}_t) \mathbf{a}_t^f}{\sum (1 - \mathbf{m}_t) \mathbf{a}_t^f}, \tag{2}$$

where $\mathbf{m}_t$ and $\mathbf{a}_t$ are again the match vector and the attention weight vector, respectively, now combined by element-by-element multiplication, and $f$ is a free attentional focusing parameter. When the outcome of the current trial is correct, then the feedback signal is computed only with the matching attention weights, when the outcome is incorrect, only the non-matching attention weights contribute to the feedback signal.

Finally, the attention weights are updated proportionally to the weighted feedback signal:

$$\begin{aligned} \mathbf{a}_{t+1} | \text{right} = (1 - r)\mathbf{a}_t + r\mathbf{s}_t \\ \mathbf{a}_{t+1} | \text{wrong} = (1 - p)\mathbf{a}_t + p\mathbf{s}_t \end{aligned}, \tag{3}$$

where $r$ and $p$ (reward and punishment sensitivity) are two weighting factors (free parameters).

**Model comparison**. We compared the full Bishara model (here called RPDF) against three other degenerate versions of the model that fixed one of the four estimable parameters to test the necessity of the different parameters. The first model variant (RRDF) assumed only a single common learning rate for reward and punishment. Given the importance of the P parameter for perseverative errors and the observed difference between R and P in our model parameters, comparison to this variant showed that two different estimable learning rates are essential for modeling our data with the Bishara model.

The second model variant (RP1F) fixes the decision consistency parameter D to 1 and restricts the decision noise to a moderate level. Low values of D lead to random choices and high values of D lead to choices that are strongly driven by the differences in the attention weights. Comparison with this model variant demonstrates whether a freely varying decision consistency parameter across participants leads to a better fit of the full model, which captures the between-subject variance in choice consistency.

The third model variant (RPD0) fixes the attentional focus parameter F to 0 leading to the update of all attention weights equally. A parameter of F = 1 updates the attention weights proportional to their current values. Higher F values lead to increasing focus on only the dimension of the higher attention weight and this in turn increases perseverative errors (Supplementary Figure 4). Comparison of the full model against this variant shows whether a freely estimable F parameter leads to a better model fit, underscoring that patients with PFC lesions exhibit suboptimal attentional focusing.

We calculated the deviance information criterion (DIC) to formally compare between these different versions of the model. DIC takes accuracy of the model fit (deviance) and model complexity (effective number of parameters) into account. In Table 2 we present ΔDIC values where we subtracted the DIC score of one of the model variants from the DIC score of a randomly choosing agent (DIC_random = $-2 \times \log(0.25)$ for a 4-option choice for all trials and participants). Here, higher values indicate better model fit.

**Model estimation**. The model was fitted to the behavioral data using Bayesian estimation[65] by estimating the actual posterior distribution of the model parameters at the individual participant level. Supplementary Figure 2 shows a graphical representation of the model. Computation of the posterior was conducted using Markov Chain Monte Carlo (MCMC) sampling using the JAGS software[66]. Individual model parameters were summarized by the posterior mode (maximum a posteriori (MAP)) as a point estimate. These values were then used as inputs for the lesion mapping and cluster analysis.

**Lesion mapping analysis**. All neuroanatomical data were mapped using MAP-3[67]. Because the neuroanatomical data were manually traced by a neuroanatomical expert (Hanna Damasio) to a stereotaxic template (for details see Supplementary Note 2), no automated spatial normalization was required. We used VLSM[68] to identify the neural correlates associated with lower values of our four model parameters. We used the Brunner–Munzel test at a threshold of 1% FDR, which corresponds to a critical Z-threshold of 3.1. This test is implemented in the "Nonparametric Mapping (NPM)" tool (version 2 May 2016) that is a part of the

MRIcron software package[36] (http://www.mccauslandcenter.sc.edu/mricro/mricron/). We placed an initial lower bound on statistical power by including in all subsequent analyses only those voxels having a lesion overlap from at least 12 patients. See Supplementary Information for details.

Using multiple linear regression, we checked for the possible confounding effects of variables that might be correlated with our experimental variables of interest, specifically gender, handedness, education, overall lesion volume, and all background neuropsychological measures that are listed in Supplementary Table 2. The resulting residualized model parameters and WCST scores correlated with their original values at 0.9 or higher, indicating that the effect of all these possible confounds on the WCST is negligible. We therefore used unresidualized WCST scores in all our analyses.

**Cluster analysis**. We used a two-step $k$-means clustering on the four model parameters to identify different subgroups of patients based on their multivariate profile of model parameters. The most appropriate number of cluster and distance measures was identified by comparing the mean silhouette value (see Supplementary Table 3) of clustering solutions based on 2 to 7 clusters and their Euclidean, Cityblock, and Correlation distances. Profiles of model parameters and WCST scores were calculated for each cluster. Furthermore, three lesion density maps were computed for the patients in each cluster and thresholded at >8 lesions.

We validated our clustering solutions using the following approach. We drew 1000 random samples of 164 participants with replacement (50% of our full sample of 328) from our set of patients with lesions. Each of these samples was submitted to the same two-step cluster analysis as above, and the mean model parameters and WCST scores were computed and compared against the original profiles (Supplementary Figure 11).

**Code availability**. Custom-made MATLAB code is available upon request from the authors.

## Data availability

Data are available upon request from the authors, as permitted under HIPAA regulations.

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

## Acknowledgements

We thank Friederike Irmen, Arnina Frank, and Anne Bert for help with coding the original test sheets into a digital format. This work was supported by the German Ministry of Education and Research (Bernstein Award for Computational Neuroscience, 01GQ1006) and the German Research Foundation (SFB TRR 169 "Crossmodal Learning") to J.G., by a McDonnell Foundation Collaborative Action Award (#220020387) to D.T., by a Conte Center grant from the NIMH to R.A. and D.T. (2P50MH094258), and by the Carver Mead New Adventures Fund to R.A.

## Author contributions

J.G. and R.A. designed the research, D.T. collected the data, J.G. analyzed the data, and all authors wrote the paper.

## Additional information

**Competing interests:** The authors declare no competing interests.

