## [Peer Review File · Nature Communications]

Reviewers' comments:

Reviewer #1 (Remarks to the Author):

Gläscher et al. used a model-based approach and a voxel-based lesion mapping in 306 patients with focal lesions to uncover the processes engaged by the WCST. They found that a key component process most responsible for impaired task performance (punishment sensitivity) is significantly associated with the right frontoparietal cortex and underlying white matter.

Overall, this is an interesting study that extended previous studies and show the possibility of using model-based approaches in understanding the patterns of deficits on classical neuropsychological tasks such as the classical WCST. However, the following concerns should be addressed before publication.

Abstract

...a deficit in flexibly switching between task sets that our model reveals arises from insensitivity to punishment...

In my view, I don't think it is appropriated to use the term 'punishment', because there is no punishment in the WCST.

Introduction

P4, ...notably Barcelo and Knight 22 proposed testing choice strategies through a further separation of nonperseverative errors into "efficient" and "random" errors.

I am very interested if the model can distinguish these two types of errors. At least, the authors should add some discussion about this point.

Same page, in the last paragraph. There are many models for WCST, but the authors just used the model of Bishara et al. So I expected the authors provided some justification about their selection.

Results

The lines and the outline in Table 1 is missing?

P9, at the end of the second paragraph, the authors stated "...we obtained very similar results (Fig. S8).

It would better to briefly report the results.

Discussion

P13, ...it is suggested to add some citing that demonstrated the importance of white matter in high order cognition.

The same page, ...which demonstrated strong negative correlation between perseverative error and punishment and sensitivity (Figure 2D)...

punishment and sensitivity should be 'punishment sensitivity'. Again, I would suggest the authors to use other terms such as "negative feedback sensitivity" rather than punishment.

Methods

P16, how many male subjects? And what is the age range?

Is the paper version or software version of WCST used?

Reviewer #2 (Remarks to the Author):

The manuscript by Glasher and colleagues describes an application of the theoretical decision model to the results of Wisconsin Card Sorting Test collected on a large sample of stroke patients. This is important, as this is a very common task that provides insight into many of the real world impairments seen following brain injury. Specifically, some individuals with brain injury seem to take extreme risks, do not adequately consider consequences of their actions, and show perseveration errors. The work attempts to predict behavior by estimating four parameters suggested by Bishara et al.: attentional focusing (F), reward sensitivity (R), punishment sensitivity (P), and decision consistency (D). What is significant about this work is the authors attempt to derive anatomical predictors for these parameters using the pattern of brain injury observed in a very large dataset of individuals with stroke (n=306). Many aspects of this work are novel, in particular to this topic. I do think this work has the potential to be influential. The scope of this work is very ambitious, and the findings make clear predictions that can be investigated using convergent methods (e.g. brain stimulation in healthy adults). Therefore, the significance is high, both clinically and theoretically. The innovation of this work is also very high. In general I thought the introduction did a nice job motivating this work, highlighting significant prior work as well as the limitations to date. Despite my general enthusiasm for this work, I do have a number of comments that should be addressed. I describe these in more detail below.

In general, this manuscript is quite terse, and I fear that people who are interested in the clinical significance will find the methods daunting and hard to untangle. I do think the introduction in particular could be tailored a bit.

While this is a very promising direction, I feel that there are some critical problems in validating the model on real subjects (predicting the lesions of left-hemiparetic participants), as described below. In addition, some information regarding validation the model on simulations is missing. Another missing piece is the comparison of model performance versus prediction of lesions from the WCST scores (see below). Together, I feel that the argument of neuroscientific validity of the decision model presented by the authors is not convincing.

I think the statement "This novel approach, model-based lesion mapping," is a bit too strong a claim. The topic and aspects of the application are novel, but related approaches have been applied to other impairments, such as perception and language (<https://www.ncbi.nlm.nih.gov/pubmed/23765000> <https://www.ncbi.nlm.nih.gov/pubmed/20028714>).

The results of the parameter recovery study, presented in supplementary figures S2 and S3, are very impressive: even though the estimation process is sometimes underestimating the values of R and P, overall the estimation accuracy is very high. What is missing is the details of estimating the WCST scores. Figure 2D shows that the population averages of the five WCST scores are estimated quite accurately, but there is so much variation in both real and predicted WCST scores that the accuracy of estimation of particular WCST scores is not evident from the figure. It would be great if the authors could add the scatterplots of actual versus predicted WCST scores in the simulated datasets.

More critically, the leave-one-out procedure used to test the model on real subjects might be biased. When doing cross-validation, it is very important that no information from the test set (in this case, the left-out participant) was used to train the predictive model (the four parameters). This does not seem to be the case in the presented study: the four parameters of Bishara's model are estimated using all 306 participants, including the left-out participant. Therefore, there is a certain amount of shared information between training and test sets, which could inflate the prediction accuracies shown on Figure 5B. I concede that in general 306 individuals is a large sample, and removing any individual should not typically have much of an influence on this estimate, but one worries about any chance of leakage in this process.

I thought the "predicting lesion location from model parameters" was a real attribute. I have heard

this notion discussed before, but the execution here is compelling. However, it would be great if the authors carried the procedure of predicting lesions, but using the WCST scores instead of the model parameters (R, P, F, D). That is, the correlational distance would be computed in five-dimensional space of (PSV, NPSV, NCAT, FSET, TRSET) rather than in four-dimensional space of (R, P, D, F). If the prediction of lesions from the four model parameters is better than prediction from the five WCST scores, this could be a strong argument in favor for the neurological relevance of Bishara's model. A related point: it would be very informative if the authors could add some supplementary material showing the actual images of predicted lesion maps, say, for a small number of participants for whom the prediction worked really well.

Bottom panel of Figure 3: red and yellow areas may be due to random noise: just because a region does not reach FDR-corrected significance does not mean it is statistically different from a region which does. A better plot would be one that looks for regions which generate a statistically significant interaction effect. In general, this concern could be raised regarding all factors: since R, P, D, F are all predictive of perseveration, however it is somewhat unclear if they are all noisy measures of PSV. Further, this lack of seeking specific interaction effects and reporting statistically significant dissociations is particularly problematic in the paragraph "Overall, the lesion density map of cluster 3 had a considerably larger total volume ..."

Minor comment: I would change "Only techniques such as TMS or lesion studies" to read "Only techniques such as brain stimulation or lesion studies"

Reviewer #3 (Remarks to the Author):

Manuscript Summary:

According to the authors, WCST outcome metrics are somewhat qualitative and fail to encompass basic cognitive processes underlying normal performance or deficits in the task. Therefore, previous anatomical studies attempting to implicate brain regions in the execution of the WCST have been unsatisfactory, methodological issues set aside. Computational modeling + VLSM = localized regions of necessary brain areas for component processes of WCST. This study showed impressive correlations and degree of lesion overlap between punishment sensitivity and perseverative errors, would seem to indicate that the crucial common deficit in patients with high perseverative errors is an insensitivity to punishment/wrong answer feedback. Provides good evidence of the power of the combination of these techniques (computational modeling and lesion method), might encourage other scientists to use this combo in their respective areas of expertise.

I believe this study will appeal to a very broad audience as it will influence thinking in the field. There is potential for re-evaluating a number of current clinical assessments in terms of more basic computational processing, from executive functions to neuropsychology to clinical psychology and psychiatry most broadly. More parcellated and fine-grained cognitive data could enhance our understanding of mental processes and improve clinical assessment in terms of predicting outcomes. Good to get people thinking beyond what is clinically useful and toward what is 'cognitively correct.'

Minor Suggested improvements:

1. Should be cautious localizing low punishment sensitivity, high PSV, and high TRSET1 exclusively to right PFC given inadequate sampling of left hemisphere lesions *despite* the VLSM analysis controlling for homogenous density. Control analysis is insufficient due to the poor sampling of posterior left hemispheric white matter; the extent/volume and lesion locations sampled in right hemispheric cases exceed that of the left hemispheric cases despite being matched in number of cases in the control analysis. This is crucial because right hemispheric posterior white matter is heavily implicated in the same control analysis (Fig S8), and in every other anatomical result (Figs

3-5). While the authors do mention that the discrepancy between the number of right and left hemisphere lesions is due to the discarding of aphasic subject data, they should specifically acknowledge whether the asymmetry of these posterior regions is due to this same issue and should address this caveat in the discussion. Otherwise, may ask authors to include more non-aphasic left hemisphere posterior cases in analysis if data is available.

2. The WCST, like every cognitive operation, takes time. Performance impairment detectable by standard outcome measures or the computational model is evidence of a more severe deficit, but slowed performance in WCST may suggest more subtle impairment. Though the lesion method is traditionally considered incapable of capturing temporally sensitive information, perhaps a computational model accounting for the speed of decisions in WCST could identify such subtle impairments. Could also be a job for fMRI. In any case, might suggest as a future direction and what the potential benefit may be in the discussion.

Clarity and context: Could be more specific in Intro about how current outcome metrics fail to encompass basic cognitive processes: "For instance, perseverative errors may reflect specific deficits in A, B, C..."

Basic edits/TYPOS

Table 1: formatting issue, no mention is made of what any of the clusters 1-3 refer to until way at the end in the methods. Should preferably give a brief explanation what the clusters are and how they were generated, or at least say "see Methods."

Figure 2: should say "healthy comparison subjects" as opposed to healthy controls, standard convention since you can't control human subjects per se, authors also refer to these subjects as 'comparisons' below so should be consistent

Figure S4: lower left plot of F against D, D axis #s should be flipped; lower middle plot of F against P, P axis #s should be flipped

Figure 4: A & B labels

Text above Figure 4 refers to Figure S8, which doesn't exist, should say S7

Figure S7: 306 patients

Discussion, Cognitive processes of the WCST, p13: "(i.e. fewer perseverative errors)"

Discussion, Neuroanatomical substrates of the WCST, p13: "...between perseverative error and punishment sensitivity (Figure 2D)" NOT 'punishment and sensitivity'

p15: "Our findings..."

Reviewer #4 (Remarks to the Author):

The authors describe a study where voxel-based lesion-symptom mapping (VLSM) is used to relate behavioral impairments in a conventional Wisconsin Card Sorting Task (WCST) to brain damage in human subjects. Behavioral impairments are characterized in two ways: 1. The parameters fit by a computational model developed by Bishara et al., and 2. Conventional neuropsychological measures of task performance. Using VLSM, the authors show that damage to a large area of the right frontal lobe is related to decreased sensitivity to punishing feedback and an increase in perseverative errors. Additional analyses identify three clusters of parameter estimates that relate to somewhat different patterns of brain damage. The goal of describing the effects of brain damage using computational modeling and psychologically meaningful parameters has great

value, and the sample size of the current study should provide more than sufficient power for the authors to reach firm conclusions. While these are great strengths, the manuscript ultimately suffers from lack of clarity and several very basic oversights that significantly undermine my enthusiasm for its publication, and raise questions about the validity of the results. There are also a few major editing errors and lots of missing information about the subjects, which make it hard to draw inferences about the data.

Major concerns:

1. On page 14, the authors mention that their sample only included patients who “did not have aphasia to such an extent so as to preclude valid performance on the task.” This is provided as rationale for explaining the larger density of lesions in the right hemisphere within the current sample. However, given that these criteria may reshape the profile of patients included in the study, it is critically important to define how this criterion was defined (i.e. did patients show problems with expression or comprehension? were aphasic symptoms assessed in a separate task?), and how many patients were excluded on this basis.

The lesion density analysis that the authors provide does not convincingly bolster the claim for lateralized deficits in the WCST. The lesion map in Figure S7 still has much more power in the right hemisphere, primarily in the frontal white matter and lateral prefrontal areas where the authors find their strongest effects.

2. While the introduction mentions some of the conflicting evidence for mapping the neural basis of WCST performance (e.g. DLPFC vs ACC), the discussion section simply argues that impairments in the task are caused by right hemisphere damage and does not go much further. Given that such a large territory of the right frontal lobe (and white matter) was implicated by the VLSM analysis, it seems evident that the WCST has low specificity in mapping to a particular PFC subregion. The manuscript is missing an opportunity to discuss the functional significance of these findings, their relevance to past work, and significance to neuropsychological studies using the WCST.

3. Very little explanation is given for the computational model uses to describe subjects' behavior. The authors acknowledge this in the discussion, but do not provide any justification aside from arguing that this model makes a reasonable set of assumptions as a starting place. This rationale unfortunately misses the utility of the entire modeling enterprise. Computational models are useful precisely because they provide a quantifiable test of the assumptions about cognitive processes that underlie behavior. Without any model comparison, there is no way to externally assess the validity of these model parameters as accurate descriptors of the cognitive processes they are meant to stand in for. Given that only two of four parameters are strongly related to neuropsychological measures, there is reason to believe that the current model is actually not optimally describing subjects' behavior. I would suggest at least comparing the current model to a simpler model with a single parameter to describe learning from both positive and negative feedback. I also suggest looking to the optimal model in Niv et al. (2015) (“Reinforcement Learning in Multidimensional Environments Relies on Attention Mechanisms”), where a simpler model performed best in a task with similarities to the WCST.

4. VLSM compares patients without any matching for age, education, estimated IQ, lesion volume or other potentially relevant demographic factors that might modulate performance in neuropsychological testing, and are matched in region of interest level comparisons in more traditional lesion studies. It is thus critical to test how these factors are related to the outcomes of interest, and preferably regress out any significant factors before carrying out VLSM analysis. However, none of this information is reported in the manuscript. While the authors note that the results “suggest” no relation between lesion volume and behavioral measures, they do not test this, though it could be done very easily.

5. Several pieces of basic information are missing from the manuscript. Although there is a reference to a supplementary table with demographic information in the methods, it is not to be

found in the manuscript. There is also no information about the etiology of subjects' lesions and use of psychoactive medication such as anticonvulsants or antidepressants.

6. The use of parameter estimate clusters is interesting and novel way of analyzing lesion data. While the authors find two different patterns of impairment within the task, there is little discussion of this finding and its significance. I also found the description of the cluster-based lesion prediction method was hard to follow. In particular, I was not sure how the distances in parameter space are being used to make predictions. Also, I feel like a clinically more relevant classification problem is the pairwise comparison of clusters with each other (i.e. cluster 1 against 2, cluster 1 against 3), rather than a classification in the three dimensional space.

It would also be helpful to provide a table identifying where the most predictive voxels were located (and their coordinates in standard space), rather than the rough description provided in the manuscript. The same is true for the VLSM results.

Minor points:

1. There are several points where figures in the supplement appear out of order. While Figure S5 is referenced first, it appears as the fifth figure. The cluster and lesion density analyses also appear in reverse order to their references in the manuscript.

2. What are the error bars in Figure 2C? These data would also be more informatively presented as scatterplots (showing the relationship of estimated neuropsychological scores with scores from simulated data).

3. The term 'model based lesion mapping' is misleading in that it suggests that a model is being used to map a lesion, rather than mapping model parameters to lesion damage (as is the case here).

4. The word 'symptom' is missing from 'voxel based lesion symptom mapping' in the abstract.

5. On page 3, the word 'requires' is used four times in two sentences.

6. In Table 1, it is not clear what the 'clusters' refer to, as this analysis is not described until much further along in the manuscript.

7. Figure S2: 'show' should be 'shown.'

8. Figure S5: 'complete' should be 'completed.'

9. Figure S7 is referenced as Figure S8. There is also a typo in the caption of S7 ("VSLM").

NCOMMS-17-21473 Response to Reviews

We are grateful to the reviewers for the very helpful feedback, and to the editor for giving us the opportunity to respond to the critiques with a revision. We have added new data from additional participants, carried out several additional analyses, and revised the framing, interpretation, and background for our paper considerably in light of the reviews. Below we respond to each reviewer's comments in detail, but first we provide a general summary of the additional analyses and changes that we made to the manuscript.

1. We added 22 new subjects
2. We tested 3 further computational models, which were degenerate versions of the original one, and estimate model parameters for each subject individually to ensure that these were unique to each subject and did not have any information leakage from the other subjects (as they could in a hierarchical model). Thus, these model parameters can be used in a cross-validation analysis (see prediction analysis).
3. We re-ran all previous VLSM analyses and also conducted one additional follow-up analysis, in which we removed covariates (demographic, neuroanatomical, other neuropsychological variables) from the model parameters and WCST scores.
4. We recomputed the k-means cluster analyses, which resulted in a 2-stage procedure.
5. Based on this new clustering solution we recomputed the prediction analysis.
6. We added a new table with demographic, neuroanatomical (lesion volume), and other neuropsychological background information
7. We created another table which lists the location of our strongest effects in standard MNI space as a reference for other researchers and clinicians
8. We added a new supplementary figure comparing the actual lesions of exemplary subjects with the predictions from our prediction analysis (Figure S12)
9. We thoroughly revised introduction and discussion, and addressed all major and minor comments from the reviewers.
10. In response to the comments we created several new figures, which are now included in the Supplemental Information

Reviewer #1 (Remarks to the Author):

Gläscher et al. used a model-based approach and a voxel-based lesion mapping in 306 patients with focal lesions to uncover the processes engaged by the WCST. They found that a key component process most responsible for impaired task performance (punishment sensitivity) is significantly associated with the right frontoparietal cortex and underlying white matter.

Overall, this is an interesting study that extended previous studies and show the possibility of using model-based approaches in understanding the patterns of deficits on classical neuropsychological tasks such as the classical WCST. However, the following concerns should be addressed before publication.

1. Abstract

...a deficit in flexibly switching between task sets that our model reveals arises from insensitivity to punishment... In my view, I don't think it is appropriated to use the term 'punishment', because there is no punishment in the WCST.

The reviewer raises a good point, since indeed the feedback in the WCST is different from some other commonly used forms of punishment in cognitive neuroscience studies

(e.g., electric shock, monetary loss). However, we do feel strongly that our administration of the WCST indeed includes punishment—it is in the form of social punishment. Such outcomes can be as aversive (or appetitive) as other forms of punishment or reward, and have been shown to recruit similar brain regions (e.g., Kohls et al. *Neuropsychologia* 51: 2062; Izuma et al. *Neuron* 58: 284; Lin et al., *Frontiers in Human Neuroscience* 6: 143). In particular, our administration of the WCST was not done on a computer, but while sitting in front of a live person, who gave the feedback “correct”, or “incorrect”. That “incorrect” serves as an instrumental form of punishment that reinforces subsequent behavior, is borne out by the fact that healthy individuals switch their behavior accordingly. In addition, Bishara et al. also use the terms “punishment sensitivity” for their P parameter. Thus, in keeping with this usage and common usage in the literature, we would prefer to retain the term “punishment” and “punishment sensitivity”. We have added a brief explanation to this effect in our revised paper.

2. Introduction

P4, ...notably Barcelo and Knight 22 proposed testing choice strategies through a further separation of nonperseverative errors into “efficient” and “random” errors.

I am very interested if the model can distinguish these two types of errors. At least, the authors should add some discussion about this point.

The reviewer raises an interesting point. However, the Bishara model indeed does not discriminate between “efficient” and “random” non-perseverative errors. While we acknowledge that this differentiation of non-perseverative errors reflects different cognitive processes, we nonetheless chose a model that was in line with the conventional WCST scoring technique, because one of our aims was to compare computational modeling of the WCST standard scores.

3. Same page, in the last paragraph. There are many models for WCST, but the authors just used the model of Bishara et al. So I expected the authors provided some justification about their selection.

We agree that there are other models; however the computational model of Bishara et al. is the simplest and yet flexible computational model that captures the standard scoring of the WCST, and that provides parameters corresponding to intuitive psychological processes. To further test the adequacy of the Bishara et al. model, we compared it with three other models, in which a parameter in the original model was fixed. The results from this comparison are now given in a new Table 2 in the revised manuscript, and clearly show that the original model provides a better fit than any of these degenerate versions.

With respect to other models, of note is the paper by Niv et al. (2015), who compared a number of different computational models ranging from naïve RL to ideal Bayesian learning on a task that shares key features with the WCST, but also includes probabilistic feedback, which is not part of the WCST. In their exploration of the model space, a feature RL with a forgetting term yielded the best fit to the data and was about as good as a model that combined RL and Bayesian learning by computing attention weights for the dimensions and features therein.

The model in the present paper (the Bishara model) resembles most closely the Hybrid Bayesian-RL model, in that it also computes and updates attention weights for all dimensions weighted by an “attentional focus” parameter (F). However, the Bishara model does not compute weights for each of the features (e.g. the different colors, numbers, and forms in the WCST), but only operates on the dimensions, which is consistent with the task instructions of the WCST. In addition, the Bishara model extends the models tested by Niv et al. by introducing different learning rates for rewards and

punishments, which our study finds to be important for explaining impairments in perseverative errors, one of the hallmarks of the WCST.

We therefore believe that the present Bishara model already incorporates several aspects that were explored in Niv et al. (2015) and were found to be important for solving hierarchical tasks such as the WCST or the Niv task. As explained above we fitted additional degenerate versions of the Bishara model to test whether all of the aspects of this model are beneficial for fitting the model to our WCST data, and found this confirmed. All of these considerations and results strongly argue that the Bishara model is indeed a very well justified model for the WCST: it captures the relevant structure of the task parsimoniously, and any reduction of it (by fixing parameters to produce degenerate models) fits the data less well than the original model.

We have also added a paragraph to the Discussion on this topic, noting the importance of future studies that compare among a larger class of models:

“Also problematic is the nature of the computational model. As the first several sections of our Results demonstrate, we were careful to ensure that the model is well behaved, that it can capture a large range of performance, and that it can regenerate standard WCST scores well. But we did not test any alternative model. It is quite conceivable that models predicated on other processes, or models with considerably more complexity, could perform as well or better. The reason that we chose the model of Bishara et al. (2010) in the first place was that it is prima facie very plausible, and parsimonious. While there are certainly an unbounded number of more complex models, it is difficult to imagine ones that are much simpler and yet still capture the basic structure of the WCST – indeed, reducing the model to three instead of four free parameters greatly reduced model-fit in our study (cf. Table 2). The model should thus be thought of as the best starting point, with possible elaborations once additional evidence supporting such elaborations would emerge.”

Results

4. The lines and the outline in Table 1 is missing?

We have changed Table 1 considerably and now include demographic information and scores of other neuropsychological tests in addition to the volumetric information for the 5 clusters that we identified in our extended sample of 328 patients. We formatted the table in a generic style, because the final layout of the table is determined by the publisher, which will impose the journal’s table formatting style.

5. P9, at the end of the second paragraph, the authors stated “...we obtained very similar results (Fig. S8). It would better to briefly report the results.

We added the following in our revised paper:

When we carried out the same analyses on this subset of patients, we obtained very similar results (Fig. S7): perseverative errors and punishment sensitivity also overlapped to a large degree in the right dorsolateral prefrontal cortex and its underlying white matter.

6. Discussion

P13, ...it is suggested to add some citing that demonstrated the importance of white matter in high order cognition.

We have added several citations of paper that clearly demonstrate the importance of white matter, across several cognitive processes: Gläscher et al., PNAS, 2010; Sutterer et al., Cortex (2016); Philippi et al., Journal of Neuroscience (2009).

7. The same page, ...which demonstrated strong negative correlation between perseverative error and punishment and sensitivity (Figure 2D)...punishment and sensitivity should be 'punishment sensitivity'. Again, I would suggest the authors to use other terms such as "negative feedback sensitivity" rather than punishment.

Thanks for spotting this typo. We have changed the text accordingly

8. Methods

P16, how many male subjects? And what is the age range?

We apologize for not including demographic information about our sample in the initial version of manuscript. We now provide this information in Table 1. We included this information in the general sample description in the Method Section (Subsection "Subjects"). However, Table 1 further breaks the demographic and volumetric information down by the different clusters of participants that we identified during our analyses.

9. Is the paper version or software version of WCST used?

The 128-item paper version was administered to the subjects, as they interacted with a live clinician doing the testing (who was blind to all aspects of our study). This information is now included in the Methods section (subsection "Wisconsin Card Sorting Test").

Reviewer #2 (Remarks to the Author):

The manuscript by Glascher and colleagues describes an application of the theoretical decision model to the results of Wisconsin Card Sorting Test collected on a large sample of stroke patients. This is important, as this is a very common task that provides insight into many of the real world impairments seen following brain injury. Specifically, some individuals with brain injury seem to take extreme risks, do not adequately consider consequences of their actions, and show perseveration errors. The work attempts to predict behavior by estimating four parameters suggested by Bishara et al.: attentional focusing (F), reward sensitivity (R), punishment sensitivity (P), and decision consistency (D). What is significant about this work is the authors attempt to derive anatomical predictors for these parameters using the pattern of brain injury observed in a very large dataset of individuals with stroke (n=306). Many aspects of this work are novel, in particular to this topic. I do think this work has the potential to be influential. The scope of this work is very ambitious, and the findings make clear predictions that can be investigated using convergent methods (e.g. brain stimulation in healthy adults). Therefore, the significance is high, both clinically and theoretically. The innovation of this work is also very high. In general I thought the introduction did a nice job motivating this work, highlighting significant prior work as well as the limitations to date. Despite my general enthusiasm for this work, I do have a number of comments that should be addressed. I describe these in more detail below.

1. In general, this manuscript is quite terse, and I fear that people who are interested in the clinical significance will find the methods daunting and hard to untangle. I do think the introduction in particular could be tailored a bit.

We have revised several sections of the paper in order to spell out the results and their significance in more detail. In particular, we have added more about the broader clinical value of this kind of approach in the introduction, where we now write:

“A clinical goal in the use of these tests is to provide both sensitivity to brain dysfunction, and specificity to particular types of brain dysfunction, by serving as markers of particular cognitive processes that are engaged by the tasks. However, this goal is challenging, and alternative analyses of the tasks into the constituent cognitive processes are rarely undertaken. An aim of the present study was to provide such a decomposition into processes that correspond to the parameters of a computational model, and to test whether such parameters might provide insight into different subtypes of frontal lobe damage that could help diagnosis. The clinical relevance of such an approach could be considerable, given that frontal-lobe dysfunction from traumatic brain injury (TBI) is a leading cause of disability in both the young and old, with an estimated 5.3 million people living with TBI-related disability in America (CDC, 2010).”

While this is a very promising direction, I feel that there are some critical problems in validating the model on real subjects (predicting the lesions of left-put participants), as described below. In addition, some information regarding validation the model on simulations is missing. Another missing piece is the comparison of model performance versus prediction of lesions from the WCST scores (see below). Together, I feel that the argument of neuroscientific validity of the decision model presented by the authors is not convincing.

2. I think the statement "This novel approach, model-based lesion mapping," is a bit too strong a claim. The topic and aspects of the application are novel, but related approaches have been applied to other impairments, such as perception and language (<https://www.ncbi.nlm.nih.gov/pubmed/23765000> <https://www.ncbi.nlm.nih.gov/pubmed/2>

0028714).

We thank the reviewer for pointing us to these papers. We have changed “This novel approach ...” to “Our approach ...” and cite the papers in our discussion of the model-based lesion mapping approach.

3. The results of the parameter recovery study, presented in supplementary figures S2 and S3, are very impressive: even though the estimation process is sometimes underestimating the values of R and P, overall the estimation accuracy is very high. What is missing is the details of estimating the WCST scores. Figure 2D shows that the population averages of the five WCST scores are estimated quite accurately, but there is so much variation in both real and predicted WCST scores that the accuracy of estimation of particular WCST scores is not evident from the figure. It would be great if the authors could add the scatterplots of actual versus predicted WCST scores in the simulated datasets.

We thank the reviewer for this excellent idea. In the revised manuscript we have now created a new figure (Figure 3) to accommodate the reviewer’s suggestion. We now show scatter plots for predicted vs. observed WCST scores along with their respective means and standard deviations in the same plot.

4. More critically, the leave-one-out procedure used to test the model on real subjects might be biased. When doing cross-validation, it is very important that no information from the test set (in this case, the left-out participant) was used to train the predictive model (the four parameters). This does not seem to be the case in the presented study: the four parameters of Bishara’s model are estimated using all 306 participants, including the left-out participant. Therefore, there is a certain amount of shared information between training and test sets, which could inflate the prediction accuracies shown on Figure 5B. I concede that in general 306 individuals is a large sample, and removing any individual should not typically have much of an influence on this estimate, but one worries about any chance of leakage in this process.

The reviewer’s concerns are valid, but we also agree with his assessment that the potential inflation of prediction accuracy is most likely negligible in a leave-one-out cross validation analysis with 328 patients (our new sample size). Nevertheless, to convince the reviewer, we have employed a slightly different model estimation procedure: we now estimate the model parameters for each individual patient by him- or herself, i.e. without the hierarchical group distribution from which the individual parameters were sampled in the prior version of the manuscript. This change was necessary, because estimating the full hierarchical model takes a long time (about 1.5 weeks) and repeating this for 328 times with one subject missing in each of them is computationally infeasible. We therefore decided to address the reviewer’s concern by estimating the model parameters for each patient individually, thus obtaining parameter estimates that are guaranteed to have no information leakage from any of the other patients in the group. These estimates can therefore be safely used in a leave-one-out cross validation analysis without potentially inflating prediction accuracies.

To validate that these individually obtained parameter estimates were not hugely different from the parameters obtained in the hierarchical estimation procedure we correlated the individual and hierarchical fits and obtained the following scatter plots and correlation coefficients:

While the correlation for the parameters P and D was reasonably high and the marginal distributions were fairly similar, both estimation procedures varied for the parameters R and F. For the R parameter, the hierarchical fit tended to produce higher parameter values with a notable “shrinkage” toward 1. For the F parameter, the hierarchical procedure exhibited shrinkage by the group distribution toward 0. This demonstrates the regularizing influence of the group distribution, as the marginal distribution of the R parameter under the hierarchical fit is clearly biased toward 1, whereas the R parameter under the individual fit is evenly distributed. This can be seen as a “distortion” of parameter values by hierarchical regularization. Of course, individually fitted parameters are not the ground truth either, but in the case where individual fits of the R parameter are close to 0 and the corresponding hierarchical fits are close to 1 this clearly shows the distortion by the overarching group distribution. Overall, we are confident that the individual fits yielded better estimates than the hierarchical procedure. In addition, as mentioned above, individual fits were only informed by the individual subject and not by the group rendering these parameter value suitable for leave-one-out cross validation.

We therefore recomputed our prediction analysis for the newly identified clusters with these individual parameter estimates and obtained very similar prediction accuracies as before (Figure 7). In addition, we compared prediction accuracies based on the distance measure from just the subjects within each cluster (colored lines) and the accuracies obtained from predictions from the rest of the sample (gray dashed lines). While both accuracies increased with higher thresholds on the prediction maps (and converged for the highest predictive voxels beyond the 90th percentile of the prediction map (Figure 7)), the one based on the subjects within each cluster was generally higher, suggesting that lesions can be more accurately predicted from subjects with similar parameter profiles.

5. I thought the "predicting lesion location from model parameters" was a real attribute. I have heard this notion discussed before, but the execution here is compelling. However, it would be great if the authors carried the procedure of predicting lesions, but using the WCST scores instead of the model parameters (R, P, D, F). That is, the correlational

distance would be computed in five-dimensional space of (PSV, NPSV, NCAT, FSET, TRSET) rather than in four-dimensional space of (R, P, D, F). If the prediction of lesions from the four model parameters is better than prediction from the five WCST scores, this could be a strong argument in favor for the neurological relevance of Bishara's model.

This is a good suggestion. We conducted the analysis that the review suggested and plot the prediction accuracies here side-by-side.

As can be seen from the figure, prediction based on the WCST scores yields almost identical accuracies as the ones based on the model parameters. This was quite surprising to us given that the clusters of the subjects were defined from the model parameters specifically, rather than the WCST scores. Our interpretation of this is that the model is in fact predicting the WCST standard scores quite accurately (see Figure 3) and is thus an adequate model for the WCST. Two improvements of the model parameter space over the WCST score space are: (1) the model parameters more closely track putative psychological processes than the WCST scores, which are likely composite, and (2) the model parameter space is in fact lower-dimensional (4 vs 5) thus giving it an advantage simply from a dimensionality reduction perspective.

6. A related point: it would be very informative if the authors could add some supplementary material showing the actual images of predicted lesion maps, say, for a small number of participants for whom the prediction worked really well.

This is a good idea. We have selected 3 representative subjects from each of the clusters 2a-c, which either had the highest, the lowest or the median predictive accuracy at the 90th percentile threshold of the prediction map. For each subject we show the individual lesion map in red, the prediction map (thresholded at 90th percentile) and the overlap in yellow. As can be seen from these images, the subjects with the highest predictive accuracy had the largest amount of overlap between their lesion and the prediction map (yellow areas) and the least amount of misprediction (green areas). This figure is now included in the Supplement as Figure S12, which we reproduce below.

Figure S12. Examples of individual lesion masks (in red) and their corresponding prediction maps (in green) thresholded at the 90th percentile. We selected the subjects with the highest, median, and lowest prediction accuracy from cluster 2a-c. Yellow regions indicate successful prediction of lesion location, green regions indicate mispredictions, red regions are areas of failed predictions of lesion locations.

7. Bottom panel of Figure 3: red and yellow areas may be due to random noise: just because a region does not reach FDR-corrected significance does not mean it is statistically different from a region which does. A better plot would be one that looks for regions which generate a statistically significant interaction effect.

We believe there might have been a miscommunication on our side regarding the interpretation of the bottom panel of the old Figure 3 (now Figure 4 in red, blue and magenta). We intended to show *only descriptively* the large overlap (in yellow) between the significant VLSM findings for punishment sensitivity (P in green) and perseverative errors (PSV, in red). This is analogous to a conjunction analysis in fMRI (using the “Conjunction Null Hypothesis” of Nichols et al., NIMG, 2005). There is no interaction between P and PSV, not in the data nor in the design.

We did *not* intend, nor did we make the claim that voxels in red (only PSV) are statistically different from voxels in yellow (the conjunction of PSV and P). However, we now make it explicit in the manuscript that the overlap is a descriptive visualization of the overlap and not another statistical analysis

8. In general, this concern could be raised regarding all factors: since R, P, D, F are all predictive of perseveration, however it is somewhat unclear if they are all noisy measures of PSV.

We thank the reviewer for raising this question. We believe that indeed they are NOT simply noisy measures of PSV for the following reason. The correlations between model parameters and WCST scores (see Figure 2) reveal that only P and D exhibit meaningful correlation coefficients with PSV and hence any predictive relationship; R and F do not correlate with PSV.

9. Further, this lack of seeking specific interaction effects and reporting statistically significant dissociations is particularly problematic in the paragraph "Overall, the lesion density map of cluster 3 had a considerably larger total volume ..."

We thank the reviewer for this point, which has been addressed by a larger subject sample with novel analyses. Our extended sample is now 328 patients, and it yielded a new 2-step cluster solution, which of course also resulted in different lesion volumes for each cluster (see new Table 1). Therefore, this section presenting the volumetric results has been completely rewritten.

10. Minor comment: I would change "Only techniques such as TMS or lesion studies" to read "Only techniques such as brain stimulation or lesion studies"

The manuscript has been changed accordingly.

Reviewer #3 (Remarks to the Author):

Manuscript Summary:

According to the authors, WCST outcome metrics are somewhat qualitative and fail to encompass basic cognitive processes underlying normal performance or deficits in the task. Therefore, previous anatomical studies attempting to implicate brain regions in the execution of the WCST have been unsatisfactory, methodological issues set aside. Computational modeling + VLSM = localized regions of necessary brain areas for component processes of WCST. This study showed impressive correlations and degree of lesion overlap between punishment sensitivity and perseverative errors, would seem to indicate that the crucial common deficit in patients with high perseverative errors is an insensitivity to punishment/wrong answer feedback. Provides good evidence of the power of the combination of these techniques (computational modeling and lesion method), might encourage other scientists to use this combo in their respective areas of expertise.

I believe this study will appeal to a very broad audience as it will influence thinking in the field. There is potential for re-evaluating a number of current clinical assessments in terms of more basic computational processing, from executive functions to neuropsychology to clinical psychology and psychiatry most broadly. More parcellated and fine-grained cognitive data could enhance our understanding of mental processes and improve clinical assessment in terms of predicting outcomes. Good to get people thinking beyond what is clinically useful and toward what is 'cognitively correct.'

Minor Suggested improvements:

1. Should be cautious localizing low punishment sensitivity, high PSV, and high TRSET1 exclusively to right PFC given inadequate sampling of left hemisphere lesions *despite* the VLSM analysis controlling for homogenous density. Control analysis is insufficient due to the poor sampling of posterior left hemispheric white matter; the extent/volume and lesion locations sampled in right hemispheric cases exceed that of the left hemispheric cases despite being matched in number of cases in the control analysis. This is crucial because right hemispheric posterior white matter is heavily implicated in the same control analysis (Fig S8), and in every other anatomical result (Figs 3-5). While the authors do mention that the discrepancy between the number of right and left hemisphere lesions is due to the discarding of aphasic subject data, they should specifically acknowledge whether the asymmetry of these posterior regions is due to this same issue and should address this caveat in the discussion. Otherwise, may ask authors to include more non-aphasic left hemisphere posterior cases in analysis if data is available.

We acknowledge the concerns that the reviewer raises regarding the exclusion of aphasic patients with left hemispheric lesions, which resulted in an imbalanced lesion density map especially in the posterior parts of the brain. This was the reason why left posterior regions were not included in the VLSM analysis (Figure 4), because we did not have sufficient statistical power in these regions. Nevertheless, the reviewer raises the question of how left posterior subjects would perform compared to right posterior subjects, which are implicated in WCST impairments.

To address this issue we identified 13 aphasic subjects with left posterior lesions in our sample who nonetheless gave valid WCST performances. These subjects were already part of the original sample in the initial version of the manuscript. We compared their WCST scores and model parameters to those from patients with right posterior lesions. We identified the latter group of patients by flipping the density map of the aphasic subjects into the right hemisphere and searching for subjects whose lesions overlapped with this flipped lesion mask. This resulted in 27 subjects with right posterior lesions. Their WCST scores and model parameters were remarkably similar to those from the aphasic patients (see figure below).

These findings suggest that the reviewer appears to be correct in cautioning us to place too much emphasis on right anterior and posterior involvement in PSV error and punishment sensitivity. Apparently, left posterior lesions may also cause a similar behavioral and cognitive profile as right posterior lesions, We have described this additional comparison in the Results and have changed the Discussion accordingly. The figure below is now included in the Supplement as Figure S8

Figure S8. Comparison of profiles of model parameters and WCST scores for 13 aphasic patients with left posterior lesions and 27 patients with right posterior lesions.

2. The WCST, like every cognitive operation, takes time. Performance impairment detectable by standard outcome measures or the computational model is evidence of a more severe deficit, but slowed performance in WCST may suggest more subtle impairment. Though the lesion method is traditionally considered incapable of capturing temporally sensitive information, perhaps a computational model accounting for the speed of decisions in WCST could identify such subtle impairments. Could also be a job for fMRI. In any case, might suggest as a future direction and what the potential benefit may be in the discussion.

We thank the reviewer for this insightful suggestion. However, we do not have the reaction time data to address the issue of slowed response times in the WCST. Therefore, any computational model that would propose a temporally specific effect would be completely hypothetical (not validated by the data) and would rest only on the assumption of the model. We are therefore unfortunately unable to carry out this interesting suggestion—we just don't have the data. We now explicitly acknowledge this important issue, and suggest it as an important future direction in our Discussion section.

Clarity and context: Could be more specific in Intro about how current outcome metrics fail to encompass basic cognitive processes: "For instance, perseverative errors may reflect specific deficits in A, B, C..."

Thank you for pointing this out. We now mention that perseverative errors can result from deficient reward/punishment processing or from a lack of cognitive flexibility, and more generally how the scores from standard clinical tasks likely reflect multiple cognitive processes that are important to disentangle—hence the value of such modeling (see also point 5 from Reviewer #2).

Basic edits/TYPOS

Table 1: formatting issue, no mention is made of what any of the clusters 1-3 refer to until way at the end in the methods. Should preferably give a brief explanation what the clusters are and how they were generated, or at least say “see Methods.”

We now have an outline of all the analyses at the beginning of the Results and there we also mention the cluster analysis.

Figure 2: should say “healthy comparison subjects” as opposed to healthy controls, standard convention since you can’t control human subjects per se, authors also refer to these subjects as ‘comparisons’ below so should be consistent

The text was changed accordingly.

Figure S4: lower left plot of F against D, D axis #s should be flipped; lower middle plot of F against P, P axis #s should be flipped

Thanks for your careful inspection of this figure. We have also checked the data and the plots again and have concluded that the numbers in the F against P plot are correct (you get fewer PSV errors with large P parameter). The unusual alignment of the axis labels is a result of our choice of aspect ratio in which we tried to make the plane in the plot most visible. However, for the F against D plot we have changed the orientation so that the increase in the F and D axis is now more intuitive.

Figure 4: A & B labels

Text above Figure 4 refers to Figure S8, which doesn’t exist, should say S7

Figure S7: 306 patients

Discussion, Cognitive processes of the WCST, p13: “(i.e. fewer perseverative errors)”

Discussion, Neuroanatomical substrates of the WCST, p13: “...between perseverative error and punishment sensitivity (Figure 2D)” NOT ‘punishment and sensitivity’

p15: “Our findings...”

All the typos have been fixed in the revised manuscript. The numbering of the figures and supplemental figures has changed and the erroneous references have been fixed. We thank the reviewer for this careful and helpful reading of our manuscript.

Reviewer #4 (Remarks to the Author):

The authors describe a study where voxel-based lesion-symptom mapping (VLSM) is used to relate behavioral impairments in a conventional Wisconsin Card Sorting Task (WCST) to brain damage in human subjects. Behavioral impairments are characterized in two ways: 1. The parameters fit by a computational model developed by Bishara et al., and 2. Conventional neuropsychological measures of task performance. Using VLSM, the authors show that damage to a large area of the right frontal lobe is related to decreased sensitivity to punishing feedback and an increase in perseverative errors. Additional analyses identify three clusters of parameter estimates that relate to somewhat different patterns of brain damage. The goal of describing the effects of brain damage using computational modeling and psychologically meaningful parameters has great value, and the sample size of the current study should provide more than sufficient power for the authors to reach firm conclusions. While these are great strengths, the manuscript ultimately suffers from lack of clarity and several very basic oversights that significantly undermine my enthusiasm for its publication, and raise questions about the validity of the results. There are also a few major editing errors and lots of missing information about the subjects, which make it hard to draw inferences about the data.

Major concerns:

1. On page 14, the authors mention that their sample only included patients who “did not have aphasia to such an extent so as to preclude valid performance on the task.” This is provided as rationale for explaining the larger density of lesions in the right hemisphere within the current sample. However, given that these criteria may reshape the profile of patients included in the study, it is critically important to define how this criterion was defined (i.e. did patients show problems with expression or comprehension? were aphasic symptoms assessed in a separate task?), and how many patients were excluded on this basis.

We now address this issue in three ways. First, we recruited an additional 25 subjects, including several with aphasia. However, we excluded 3 patients based on our thresholds for excluding patients with severe aphasia (see revised Methods). This resulted in 328 subjects in the new sample (306 from the original sample and 22 new subjects). Second, we conducted an explicit test of patients with lesions in left and right posterior cortex, which is given in the new Supplemental Figure S8 (please see response to Reviewer #3, point 1, above). Third, we provide additional details in the revised methods on how aphasic participants were excluded. The summary of this is that they were excluded on the basis of an independent task commonly used to assess the ability to comprehend basic syntax, the Token Test.

The lesion density analysis that the authors provide does not convincingly bolster the claim for lateralized deficits in the WCST. The lesion map in Figure S7 still has much more power in the right hemisphere, primarily in the frontal white matter and lateral prefrontal areas where the authors find their strongest effects.

We appreciate and share the concern of the reviewer, which is the reason why we undertook the equalized lesion density analysis in Figure S7 in the first place. Reviewer 3 raised the same issue w.r.t. strong claims about the right posterior involvement in PSV and P.

Nevertheless, it is not always possible to equalize lesion density across the entire brain and still have enough statistical power to draw conclusions given the exclusion criteria above. As noted above, we addressed this issue by inspecting the profiles of 13 aphasic subjects who had mostly left posterior lesions. By flipping their density map into the right hemisphere we were able to identify 27 subjects with right posterior lesions in those contralateral sectors, thus providing a qualitative comparison between the effects of homotopic regions lesions. We compared the profiles of model parameters and WCST

scores between these two groups. (Please see Figure S8 and our response to Reviewer 3). Both groups of lesion patients had almost identical profiles suggesting (a) that specific impairments in WCST performance are not exclusive to the right posterior cortex, and (b) that aphasic subjects were still capable of completing the WCST in the same way as non-aphasic subjects. The caveat to this analysis is that we did not have enough lesions in the left posterior hemisphere to make a statistical claim, and hence these findings remain qualitative. We mention this additional analysis in the revised manuscript now and are more cautious about our conclusions on right posterior involvement.

2. While the introduction mentions some of the conflicting evidence for mapping the neural basis of WCST performance (e.g. DLPFC vs ACC), the discussion section simply argues that impairments in the task are caused by right hemisphere damage and does not go much further. Given that such a large territory of the right frontal lobe (and white matter) was implicated by the VLSM analysis, it seems evident that the WCST has low specificity in mapping to a particular PFC subregion. The manuscript is missing an opportunity to discuss the functional significance of these findings, their relevance to past work, and significance to neuropsychological studies using the WCST.

We thank the reviewer for pointing this out and have thoroughly revised our Discussion section. We now acknowledge that a broad region was implicated in task impairments in our study, and that white matter involvement is important. We also note that these are clinically important findings, and that further anatomical specificity is certainly possible provided larger sample sizes are accrued.

3. Very little explanation is given for the computational model uses to describe subjects' behavior. The authors acknowledge this in the discussion, but do not provide any justification aside from arguing that this model makes a reasonable set of assumptions as a starting place. This rationale unfortunately misses the utility of the entire modeling enterprise. Computational models are useful precisely because they provide a quantifiable test of the assumptions about cognitive processes that underlie behavior. Without any model comparison, there is no way to externally assess the validity of these model parameters as accurate descriptors of the cognitive processes they are meant to stand in for. Given that only two of four parameters are strongly related to neuropsychological measures, there is reason to believe that the current model is actually not optimally describing subjects' behavior. I would suggest at least comparing the current model to a simpler model with a single parameter to describe learning from both positive and negative feedback.

The reviewer's criticism about the missing model comparison is valid. We have therefore expanded our model space and now include 3 different versions of the Bishara model, in which some of the parameters have been fixed. These degenerate versions of the model are described in the revised Methods. The results of this model comparison are shown in Table 2, and clearly show that all the degenerate versions of the original model provide a poorer fit than does the original.

I also suggest looking to the optimal model in Niv et al. (2015) ("Reinforcement Learning in Multidimensional Environments Relies on Attention Mechanisms"), where a simpler model performed best in a task with similarities to the WCST.

Niv et al. (2015) compared a number of different computational models ranging from naïve RL to ideal Bayesian learning on a task that shares key features with the WCST, but also includes probabilistic feedback, which is not part of the WCST. In their exploration of the model space, a feature RL with a forgetting term yielded the best fit to the data and was about as good as a model that combined RL and Bayesian learning by computing attention weights for the dimensions and features therein.

The model in the present paper (the Bishara model) resembles most closely the Hybrid Bayesian-RL model, in that it also computes and updates attention weights for all dimensions weighted by an “attentional focus” parameter (F). However, the Bishara model does not compute weights for each of the features (e.g. the different colors, numbers, and forms in the WCST), but only operates on the dimensions, which is consistent with the task instructions of the WCST. In addition, the Bishara model extends the models tested by Niv et al. by introducing different learning rates for rewards and punishments, which our study finds to be important for explaining impairments in perseverative errors, one of the hallmarks of the WCST.

We therefore believe that the present Bishara model already incorporates several aspects that were explored in Niv et al. (2015) and were found to be important for solving hierarchical tasks such as the WCST or the Niv task. As explained above we fitted additional degenerate versions of the Bishara model to test whether all of the aspects of this model are beneficial for fitting the model to our WCST data, and found this confirmed. All of these considerations and results strongly argue that the Bishara model is indeed a very well justified model for the WCST: it captures the relevant structure of the task parsimoniously, and any reduction of it (by fixing parameters to produce degenerate models) fits the data less well than the original model.

4. VLSM compares patients without any matching for age, education, estimated IQ, lesion volume or other potentially relevant demographic factors that might modulate performance in neuropsychological testing, and are matched in region of interest level comparisons in more traditional lesion studies. It is thus critical to test how these factors are related to the outcomes of interest, and preferably regress out any significant factors before carrying out VLSM analysis. However, none of this information is reported in the manuscript. While the authors note that the results “suggest” no relation between lesion volume and behavioral measures, they do not test this, though it could be done very easily.

The reviewer is correct in pointing out that demographic and other neuropsychological and neuroanatomical variables often vary with task performance and can confound or dilute the specific effects of a particular task. We conducted a follow-up VLSM analysis in which we regressed out demographics (gender, handedness, education), total lesion volume, and all the neuropsychological background variables listed in Table 1 from our target variables (model parameters and WCST scores) by means of multiple linear regression. Correlation coefficients between these residuals and the original target variables were 0.9 or above suggesting that removing the effects of all these covariates hardly changes the original target variables. The residualized target variables were then submitted to a follow-up VLSM analysis (see Figure below). As before, we only found significant effects for the residualized model parameter P and the residualized perseverative error score (PSV), which overlapped in the right PFC and posterior frontal cortex (see Figure below). While the residualized PSV was only significant at a lower statistical threshold (5% FDR instead of 1% FDR in the original analysis), the residualized P remained significant at 1% FDR.

This suggests that while the covariates do remove parts of the variance of the original effect, as would be expected (esp. for PSV), the general pattern and localization of P and PSV did not change. We mention this follow-up analysis in the results and include it as Figure S6 in the Supplement.

Figure S6. VLSM analysis of the residuals of punishment sensitivity (P) and perseverative errors (PSV) with demographic (gender, handedness, education), neuroanatomical (lesion volume) and neuropsychological background assessment (see Table 1) removed. The residuals for P reached significance at 1% FDR, whereas the residual for PSV were significant only at the reduced threshold of 5% FDR.

5. Several pieces of basic information are missing from the manuscript. Although there is a reference to a supplementary table with demographic information in the methods, it is not to be found in the manuscript. There is also no information about the etiology of subjects' lesions and use of psychoactive medication such as anticonvulsants or antidepressants.

We apologize for not including this information in the original submission of the manuscript. We have now included comprehensive information (demographic, etiological and neuropsychological background information as well as lesion volume) in Table 1.

6. The use of parameter estimate clusters is interesting and novel way of analyzing lesion data. While the authors find two different patterns of impairment within the task, there is little discussion of this finding and its significance. I also found the description of the cluster-based lesion prediction method was hard to follow. In particular, I was not sure how the distances in parameter space are being used to make predictions.

We thank the reviewer for this comment and apologize if our original description was difficult to follow. Here we provide a more detailed description. K-means clustering requires the specification of the number of clusters and the type of distance measure to be used for the analysis. We decided to test for different numbers of clusters (2-7) and chose the most appropriate number of clusters based on the average silhouette value. The silhouette value is a measure of the distance of a point in one cluster from the neighboring clusters and it ranges from "probably in this one cluster rather than any of the others" (+1) to "not distinctly belonging to either cluster" (0) to "probably belonging to

a different cluster” (a misclassification, -1). The average silhouette value is therefore an index of how distinct a particular clustering solution is.

The other parameter of k-means clustering is the distance measure. We compared 3 commonly used distance measures in our cluster analysis: Euclidean distance, correlation distance, and Cityblock distance. These different measures are defined as:

Euclidean Distance

$$D(p,q)=\sqrt{\sum_{i=1}^n(p_i-q_i)^2}$$

where p and q are the n-element vectors of model parameters from two subjects. Here: n=4 (4 model parameters)

Correlation Distance

Correlation between p and q, where p and q are the same vectors of model parameters as above

Cityblock Distance

$$D(p,q)=\sum_{i=1}^n|p_i-q_i|$$

where p and q are just as defined above.

As an example, consider the model parameters (R,P,D,F) of 2 virtual subjects: p = [0.6, 0.2, 1.7, 0.9] and q = [0.2, 0.8, 2.4, 0.5]. The different distance measures are:

Euclidean: 1.0817

Correlation: 0.7998

Cityblock: 2.1

The point of all these measures is to express the differences between two subjects in a single consistent measure across all subjects.

Because we aimed for the most distinct clustering solution we let the silhouette values decide how many clusters and what distance measure should be used. We decided to use this two-stage clustering approach because the first cluster analysis (using the Euclidean distance and k=3, see Figure S9 and Table S1) only differentiated two small high- and low-performing groups of subjects (cluster 1 and 3) and a huge cluster with intermediate performance but no further differentiation within (cluster 2). In the second cluster analysis (using the correlation distance and k=3, see Figure S10 and Table S2) we therefore used only the subjects of cluster 2 to see if we could break this apart, and indeed found 3 sub-clusters of subjects (termed 2a, 2b, and 2c) with different extent of anterior PFC damage.

Our prediction analysis was guided by the hypothesis that similar profiles of model parameters in two subjects (i.e. smaller distance) would result in similar lesion location. In a leave-one-out cross-validation analysis, we therefore used the pair-wise distance measure to weight the lesion mask of each subject in the cluster minus the left out subject. All these weighted lesion masks were then added to create the prediction map for each subject. High values in the prediction map mean that subjects with lesions in this area have a similar profile of model parameters, whereas low values in the prediction

map indicate that subjects with lesions in this area have a different profile of model parameters.

This prediction map was then thresholded at the 50th, 70th, 90th and 99th percentile and we computed the accuracy of the prediction map w.r.t to the original lesion mask of the left-out subject. This was repeated for every subject.

Accuracies for each subject were then averaged for each cluster and the average prediction map for each cluster was also computed. These data are shown in Figure 7. Examples of the individual match and mismatch between original lesion masks and prediction maps are shown in Figure S12 as requested by Reviewer 2.

We expanded the description of the prediction analysis in the Methods section to make the different steps more explicit.

Also, I feel like a clinically more relevant classification problem is the pairwise comparison of clusters with each other (i.e. cluster 1 against 2, cluster 1 against 3), rather than a classification in the three dimensional space.

This is indeed exactly what we did. Both cluster analyses are based on the pair-wise distance between the profiles of model parameters.

If the reviewer is asking for a pair-wise comparison of the lesion density maps for each cluster, this is a different question, but also important. We therefore created difference images of the cluster density maps for cluster 2a-c (the ones with intermediate, but impaired performance and extensive PFC involvement) and show them in the figure below. This figure is included in the Supplement as Figure S11

It would also be helpful to provide a table identifying where the most predictive voxels were located (and their coordinates in standard space), rather than the rough description provided in the manuscript. The same is true for the VLSM results.

Thank you for alerting us to this issue. This certainly raises the value of our findings for other researchers and clinicians who would like to compare their data with or results. We have therefore created Table 3, in which we list peak voxels (and also other local peaks within a significant region) for our VLSM and prediction analyses in standard MNI space.

Minor points:

1. There are several points where figures in the supplement appear out of order. While Figure S5 is referenced first, it appears as the fifth figure. The cluster and lesion density analyses also appear in reverse order to their references in the manuscript.

The arrangement of Figures in the Supplement has been changed in the revised manuscript and references in the main text have been updated.

2. What are the error bars in Figure 2C? These data would also be more informatively presented as scatterplots (showing the relationship of estimated neuropsychological scores with scores from simulated data).

Reviewer 2 raised a similar issue and to honor his and your criticism we have created the new Figure 3, which plots observed vs. predicted WCST scores along with their means and SDs. We have also included the original bar plot in the figure, because this might be easier to parse for the casual reader of the paper.

3. The term ‘model based lesion mapping’ is misleading in that it suggests that a model is being used to map a lesion, rather than mapping model parameters to lesion damage (as is the case here).

We use the term “model-based lesion mapping” in the same way that we and other researchers use “model-based fMRI” (Gläscher & O’Doherty, WIRE Cog Sci, 2010). The latter is now an established analysis approach in neuroimaging research, in which a computational model is fitted to behavioral data and the internal variables are then correlated with the BOLD signal. Our approach follows this logic and we therefore think that “model-based lesion mapping” will resonate with the neuroscience community and will highlight the similarity of the approaches. We reference the use of the term “model-based fMRI” in explaining our approach, in the revised Introduction to our manuscript.

4. The word ‘symptom’ is missing from ‘voxel based lesion symptom mapping’ in the abstract.

5. On page 3, the word ‘requires’ is used four times in two sentences.

6. In Table 1, it is not clear what the ‘clusters’ refer to, as this analysis is not described until much further along in the manuscript.

7. Figure S2: ‘show’ should be ‘shown.’

8. Figure S5: ‘complete’ should be ‘completed.’

9. Figure S7 is referenced as Figure S8. There is also a typo in the caption of S7 (“VSLM”).

These issues and typos have been fixed in the revised manuscript. We thank the reviewer for this thorough and helpful reading of our manuscript.

Reviewers' comments:

Reviewer #1 (Remarks to the Author):

The authors have addressed all my concerns. Although I am not fully convinced by their justification about the using of the term 'punishment', instead of 'negative feedback', I recommend this article to be published.

Reviewer #3

(No remarks to author, but in remarks to editor, states that the authors have addressed all of his/her concerns and recommends acceptance.)

Reviewer #4 (Remarks to the Author):

In this revised manuscript, Gläscher et al. have made a substantial effort to address concerns raised in the initial review. I appreciate the work that has gone into this revision and I have found that the manuscript has been improved as a result. The testing of alternative models provides greater validity for their use of the full model that is the focus of the manuscript, and I am much more convinced of their VLSM results given that they appear to survive residualization for demographic features. While the clustering analysis and predictive model for lesion location is now much better explained, I have questions about the external validity of this approach. I have additional concerns regarding seemingly contradictory details about the inclusion criteria for patients, which add to my concerns about the rigor of this work that I raised in the last submission. I will detail these concerns below.

Major concerns:

1. The idea of using computational modeling to identify a multivariate "cognitive fingerprint" for patients is very intriguing. However, I am skeptical about the external validity of the clustering revealed by this analysis, given that the clustering solution in the revised manuscript seems to differ substantially from the original submission after the addition of 22 patients. Out-of-sample testing is helpful in arguing that these patterns of behavior have meaningful relationships with brain damage, but I wonder if these patterns are reliable enough to be useful, especially given their noisy appearance and concentration in the white matter. If the authors could show that the clusters they identify are reliable in clustering solutions using permutations of randomly selected samples with half the dataset, it would go a long way in demonstrating the generalizability of the cluster solution they identified.
2. Relatedly, I think it would also be helpful for the authors to go further into discussing the significance of the patterns of impairments identified by the model, and their relationship to brain damage. At the moment, the discussion of the significance of these distinct patterns for cognitive theory is given very little attention.
3. On p. 18 the authors write, "While our study only used subjects with focal brain lesions, and thus excluded those with traumatic brain injury...." Yet, in Table 1, eight subjects are listed as having brain damage due to "head trauma" and 11 due to "unspecified etiology." This appears to contradict the previous statement, as the authors apparently cannot verify that patients with closed head TBIs were excluded. It is not clear if the patients 'head trauma' had damage due to closed and/or penetrating brain injuries.
4. I think it is notable that patients with left and right unilateral posterior brain damage showed similar patterns of performance. The clustering analysis seems to pick up on this by lumping these patients into the same group with the worst overall performance (cluster 3). Univariate VLSM would miss such a common pattern of deficit and likely overweight areas with more regional power (i.e. the right hemisphere). Given that the clustering analysis suggests that damage to

anatomically distant areas can cause similar patterns of deficits, the results appear more in line with a network account of WCST performance. It would likely be worthwhile to use a multivariate VLSM approach (e.g. Lesymap (Pustina et al., 2017)) to test if damage-parameter relationships are better explained by damage at multiple sites than one particular set of voxels.

5. It seems that the residualized parameter estimates may be more meaningful than the uncorrected estimates that are the focus of most of the manuscript. It is intriguing that after correction, the VLSM results appear to indicate that deficits have more cortical involvement and are not so strictly related to white matter damage, which is actually more hopeful for the localization of these deficits to cortical territories than the results currently presented. While I am not asking that the authors redo all their analyses with these residualized values, I think it would be helpful at the very least to include the figure that was in the rebuttal as part of the supplement.

6. It is not entirely clear how the left posterior hemisphere damage group was created. While the manuscript mentions that they were chosen for their matching lesion density in left and right posterior brain damage (p. 10), they are labeled in Supplementary Figure 8 as 'aphasic subjects.' It would be helpful if the authors could be clearer in describing their selection criteria here.

Minor comments:

1. "Encephalitis" is misspelled in Table 1.
2. On p. 4, "perseverative error" should be plural (i.e. "errors").
3. Figure 7 caption, should say "obtained" not "obtain."

Overall Comments:

We appreciate the opportunity to improve our manuscript further, and to address the further comments from Reviewer #4. We have gone through the entire manuscript and all of the Supplemental Information carefully, and we have addressed all of the criticisms and recommendations of Reviewer #4. Also, we decided to use “participants” rather than “subjects,” as the former is more in current fashion; this change was implemented throughout.

Detailed responses (in blue) to Reviewer #4 are provided below on a point by point basis.

Reviewer #4 (Remarks to the Author):

In this revised manuscript, Gläscher et al. have made a substantial effort to address concerns raised in the initial review. I appreciate the work that has gone into this revision and I have found that the manuscript has been improved as a result. The testing of alternative models provides greater validity for their use of the full model that is the focus of the manuscript, and I am much more convinced of their VLSM results given that they appear to survive residualization for demographic features. While the clustering analysis and predictive model for lesion location is now much better explained, I have questions about the external validity of this approach. I have additional concerns regarding seemingly contradictory details about the inclusion criteria for patients, which add to my concerns about the rigor of this work that I raised in the last submission. I will detail these concerns below.

Major concerns:

1. The idea of using computational modeling to identify a multivariate “cognitive fingerprint” for patients is very intriguing. However, I am skeptical about the external validity of the clustering revealed by this analysis, given that the clustering solution in the revised manuscript seems to differ substantially from the original submission after the addition of 22 patients. Out-of-sample testing is helpful in arguing that these patterns of behavior have meaningful relationships with brain damage, but I wonder if these patterns are reliable enough to be useful, especially given their noisy appearance and concentration in the white matter. If the authors could show that the clusters they identify are reliable in clustering solutions using permutations of randomly selected samples with half the dataset, it would go a long way in demonstrating the generalizability of the cluster solution they identified.

We appreciate the Reviewer’s concern about the differences between the first clustering solution that we reported in the initial submission of the manuscript, and the current version. We tend to agree with the Reviewer that adding 22 new patients might not be expected to change the cluster solution dramatically. However, we not only added 22 new patients, but we also employed a different model estimation technique, and these revisions together account for the differences. For the original submission we used a hierarchical Bayesian model, in which individual model parameters were drawn from an overarching group distribution. In the course of model estimation, the individual model parameters inform the group posterior, which in turn also affects the subject-specific parameters. While hierarchical estimation is a commonly used and versatile approach that regularizes individual parameters, while still allowing variations between them, the resulting set of subject-specific parameters are not fully independent of each other.

In the previous review, Reviewer 2 suggested that we should run a cross-validation analysis on the prediction data (i.e. predicting lesion from cognitive profiles), which we did in our revision. This necessitates fully independent subject-specific parameters, as the left-out samples cannot (and should not) be influenced in any way by the rest of the data. Therefore, we changed model estimation from a hierarchical approach to a subject-specific estimation (still using Bayesian estimation with MCMC sampling in JAGS). This is reflected in the updated Figure S2, in which the group distributions are now omitted.

Importantly, we compared the new subject-specific parameters obtained from individual estimation to the original parameters from the hierarchical estimation and found that they mostly differed in the R, D, and F parameters (most strongly in R). This is shown in the Figure below (Fig. Author Response 1).

Figure Author Response 1. Parameter distributions compared between our original hierarchical model, and the new revised model based on individual fits.

As can be seen in this figure above, hierarchical estimation exerted a certain degree of shrinkage (regularization) toward 1 for the R parameter.

However, new individual model parameters also change the Euclidean distance measures between all subjects, which are the basis for the k-means clustering analysis. This is likely the primary reason why we obtained differences between the initial and the current cluster solutions to which the Reviewer is referring.

To address empirically the Reviewer's question about the reliability of the clustering solution, we conducted a cross-validation analysis along the lines that the Reviewer outlined. Specifically, we drew 1000 random subsamples of 164 subjects from our data (with replacement), which represents half the sample size (total $n=328$). We then ran the same 3-means cluster analysis on each of these subsets, computed the mean WCST scores and re-sorted the cluster assignment into a cluster of low, medium, and high perseverative errors (the same groups that we detected in the original cluster analysis with the full sample). This resulted in remarkably similar group sizes (Table S2):

Cluster No.		Full sample (divided by 2)	Cross-validation samples
1		36 (18) ¹	19.3
2		266 (133)	131.4
	2a	95 (47.5)	37.4
	2b	93 (46.5)	52.6
	2c	78 (39)	41.4
3		26 (13)	13.5

¹ The number in parentheses is half the of the number of subjects in the cluster to make it comparable with the cross-validation samples, which are 50% of the full sample size

Table S2 (reproduced): *Cross-validation to verify stability of clustering solution.*

Below we show the mean model parameters and WCST scores of the 1000 cross-validation samples (represented as bars in the figures below) in comparison with the parameters and WCST scores obtained in the cluster analyses with the full sample (represented as horizontal lines across each bar). These figures (for both cluster analyses) reveal that the cross-validation cluster analysis is remarkably similar to the original solution. Thus, we are confident that the cognitive profiles of model parameter and WCST scores obtained from the full sample are reliable. These figures are now included in the Supplemental Information as Figure S11

Figure S11 (reproduced). Model parameters and WCST scores of 1000 cross-validation samples.

2. Relatedly, I think it would also be helpful for the authors to go further into discussing the significance of the patterns of impairments identified by the model, and their relationship to brain damage. At the moment, the discussion of the significance of these distinct patterns for cognitive theory is given very little attention.

We appreciate the opportunity to expand on the implications of our findings for neuropsychology and cognitive neuroscience. Following the Reviewer's suggestion, we elaborated the discussion of our findings in various places throughout the manuscript, especially in the Discussion section.

However, we note that we are limited by the word count restrictions allowed by Nature Communications in the extent to which we can elaborate on this topic here.

3. On p. 18 the authors write, “While our study only used subjects with focal brain lesions, and thus excluded those with traumatic brain injury....” Yet, in Table 1, eight subjects are listed as having brain damage due to “head trauma” and 11 due to “unspecified etiology.” This appears to contradict the previous statement, as the authors apparently cannot verify that patients with closed head TBIs were excluded. It is not clear if the patients ‘head trauma’ had damage due to closed and/or penetrating brain injuries.

We apologize for this inconsistency, and we appreciate the Reviewer calling this to our attention. In fact, the 8 TBI subjects listed in Table 1 had focal contusions caused by closed head trauma, with no evidence of diffuse brain injury (per 3T MRI and neuropsychological assessment). We only enroll such patients in our Patient Registry if they meet our strict inclusion and exclusion criteria, including having a focal, stable, and quantifiable lesion, and having a focal, stable neuropsychological deficit. These 8 patients met our criteria and were therefore included in the study. The 11 unspecified etiologies were incorrectly labeled as “unspecified,” and we have fixed this (all of the 11 had one of the etiologies listed in Table 1B, mostly stroke or resection, and we added them to the appropriate groups). We modified the Method section on “Participants” to incorporate these changes, and we also modified Section B of Table 1 to incorporate these changes.

4. I think it is notable that patients with left and right unilateral posterior brain damage showed similar patterns of performance. The clustering analysis seems to pick up on this by lumping these patients into the same group with the worst overall performance (cluster 3). Univariate VLSM would miss such a common pattern of deficit and likely overweight areas with more regional power (i.e. the right hemisphere). Given that the clustering analysis suggests that damage to anatomically distant areas can cause similar patterns of deficits, the results appear more in line with a network account of WCST performance. It would likely be worthwhile to use a multivariate VLSM approach (e.g. Lesymap (Pustina et al., 2017)) to test if damage-parameter relationships are better explained by damage at multiple sites than one particular set of voxels.

We appreciate the Reviewer’s suggestion that multivariate methods could potentially pick-up on subtle similarities in lesion-deficit correlations between both hemispheres that would not be detected by univariate methods like VLSM. Indeed, canonical correlation analysis is the method of choice when aiming for the best fit in relationships between a multivariate pattern of lesions and a multivariate pattern of model parameters. In principle, this could offer a better fit between our multiple model parameters and multiple lesion sites; however, it would depend on having a sufficiently large amount of data and, importantly, it would also forego information about which specific model parameters are best associated with which specific anatomical regions. The latter may be of more value clinically, although a CCA approach remains scientifically intriguing.

Given the scope of our paper, we decided not to add the Lesymap approach, since we feel that considerable further analyses would then be required to fully present and decipher the results (i.e., which specific tasks are driving effects in which specific lesion locations). Also, and importantly, the impressive simulations using SCCAN that are presented in Pustina et al. (2017) were limited to a single behavioral measure. In fact, the Lesymap documentation as currently written is not capable of handling multivariate behavioral data. We had email conversations with Dr. Pustina and confirmed that the SCCAN method in its current incarnation in Lesymap is not yet set up to process multiple behavioral measures at once. The code can be rewritten for multiple behavioral measures, but this is well beyond the scope of the current paper. In short: we agree with the Reviewer that canonical correlation approaches are an interesting type of analysis to explore in future studies, but we felt that this is outside the scope of our paper, and the recommended method is not directly useable for multiple behavioral parameters simultaneously.

We do agree with the Reviewer that the commonality of the two posterior groups is interesting, and we have added more discussion of this in the relevant places in the revised manuscript.

5. It seems that the residualized parameter estimates may be more meaningful than the uncorrected estimates that are the focus of most of the manuscript. It is intriguing that after correction, the VLSM results appear to indicate that deficits have more cortical involvement and are not so strictly related to white matter damage, which is actually more hopeful for the localization of these deficits to cortical territories than the results currently presented. While I am not asking that the authors redo all their analyses with these residualized values, I think it would be helpful at the very least to include the figure that was in the rebuttal as part of the supplement.

We agree on the utility of this figure and as the reviewer requested have included it in our Supplement as Figure S6.

6. It is not entirely clear how the left posterior hemisphere damage group was created. While the manuscript mentions that they were chosen for their matching lesion density in left and right posterior brain damage (p. 10), they are labeled in Supplementary Figure 8 as ‘aphasic subjects.’ It would be helpful if the authors could be clearer in describing their selection criteria here.

We conducted this analysis in response to a concern raised by Reviewer 3 during the last round of revisions regarding differences in statistical power in the posterior part of the brain due to the exclusion of severely aphasic subjects. Specifically, we identified 13 aphasic subjects with left posterior lesions, who were carefully screened (by a neuropsychologist blind to this study’s objectives and hypotheses) to assure that they could provide valid WCST data and computed their average profile of model parameters and WCST scores. Then, we asked the question whether this left posterior profile was similar to one obtained from subjects with right posterior lesions. We identified these right posterior lesion subjects by flipping the left posterior density map onto the right hemisphere and selecting those subjects whose right-sided lesions overlapped with this flipped density map. We identified 27 subjects this way and computed their profile of model parameters and WCST scores. Both sets of profiles (from the left and right hemisphere) were very similar, suggesting that posterior involvement in perseverative errors is not specific to right posterior lesion patients. This is an important and interesting outcome, and we discuss these findings in the revised results.

We agree with the Reviewer that our label was not optimal, and we changed the label in Figure S8 from “aphasic patients” to “left posterior patients” and we also revised the subject description to read as follows:

We also excluded patients who had aphasia of such severity as to interfere with comprehension of the WCST instructions and preclude valid WCST performance. Specifically, we excluded 3 such patients, based on the dual criteria of having scores < 35 on the Token Test and scores < 15 on the MAE Aural Comprehension Test.

This information was added to the revised manuscript and supplemental materials.

Minor comments:

1. "Encephalitis" is misspelled in Table 1.
2. On p. 4, "perseverative error" should be plural (i.e. "errors").
3. Figure 7 caption, should say "obtained" not "obtain."

These errors have been corrected.

Reviewers' comments:

Reviewer #4 (Remarks to the Author):

In this second round of revision, Glascher, Adolphs and Tranel have responded to my previous concerns regarding the clustering analysis, lesion-symptom mapping and lesion etiologies. I am satisfied with their response to these points, and I find that the discussion of the manuscript is significantly improved. However, in re-reading the manuscript, I have noticed some problems that I did not catch in my last review. I apologize for drawing out the review process in this way, but I believe these are important points that need to be addressed before this manuscript can be published. I will describe these below.

1. I am afraid that there is a serious issue with the cross-validation procedure described in the prediction analysis section. As I understand it, the authors calculated the correlation distance for each patient in the sample with a left-out patient. These correlation distances were then used to weight the lesion masks in the sample, which were then summed to create a predictive map, where each voxel has a weight that reflects its relation of damage to this voxel in parameter space the left out subject. The authors then test if this map is predictive of lesion location by thresholding this map and comparing it against the lesion mask of the left-out patient. This is fundamentally not a legitimate cross-validation procedure because the weights for the predictive model are based on the left-out test data. It is thus unsurprising that the predictive accuracies are fairly high. If the authors want to pursue this analysis, they need to find a way to create a predictive model that does not use the left-out data in the generation of weights whatsoever, or remove this section of the manuscript.

2. I was also asked by Dr. Horder to respond to a comment made by Reviewer 2 regarding another potential for bias in the predictive analysis. That reviewer pointed out that a hierarchical estimation procedure could create a bias in the cross-validation procedure because all parameter estimates were regularized with respect to a group mean. The authors responded in the last revision by changing their analysis to fit the model to each subject individually. I believe that this change should sufficiently remove the source of bias referenced by Reviewer 2.

3. In supplementary figures 3 and 4 demonstrating parameter recovery with this model, the figure legend describes the fitting as a hierarchical procedure. However, the authors are now using individual fits for the majority of their analyses since the last revision. At a quick glance, these figures look pretty much identical to those from the original submission where a hierarchical procedure was used. Given that removing the hierarchical procedure could also affect the fidelity of parameter recovery, it is important to complete this analysis with individual fits.

Minor:

1. P. 22, 'prediction analysis' 1st paragraph, the authors write that 'high distance values indicate that the profiles of model parameters between tow participants are very similar.'" However, for correlation distances, the opposite is true.

2. P. 22, 'prediction analysis' 2nd paragraph, should say 'create' instead of 'created.'

3. Orange line for healthy subject performance in Figure 2 is missing for the F parameter.

4. Typo in legend for Figure S7 — 'VLSM' instead of 'VSLM.'

5. Panel labels are missing from Figure 7.

Reviewer #4 (Remarks to the Author):

In this second round of revision, Glascher, Adolphs and Tranel have responded to my previous concerns regarding the clustering analysis, lesion-symptom mapping and lesion etiologies. I am satisfied with their response to these points, and I find that the discussion of the manuscript is significantly improved. However, in re-reading the manuscript, I have noticed some problems that I did not catch in my last review. I apologize for drawing out the review process in this way, but I believe these are important points that need to be addressed before this manuscript can be published. I will describe these below.

We thank the reviewer for taking the time to carefully review our manuscript again, hence giving us an opportunity to make further improvements to the paper. Our goals are aligned with those of the Reviewer – we want an accurate and quantitatively valid set of results, clearly presented.

1. I am afraid that there is a serious issue with the cross-validation procedure described in the prediction analysis section. As I understand it, the authors calculated the correlation distance for each patient in the sample with a left-out patient. These correlation distances were then used to weight the lesion masks in the sample, which were then summed to create a predictive map, where each voxel has a weight that reflects its relation of damage to this voxel in parameter space the left out subject. The authors then test if this map is predictive of lesion location by thresholding this map and comparing it against the lesion mask of the left-out patient. This is fundamentally not a legitimate cross-validation procedure because the weights for the predictive model are based on the left-out test data. It is thus unsurprising that the predictive accuracies are fairly high. If the authors want to pursue this analysis, they need to find a way to create a predictive model that does not use the left-out data in the generation of weights whatsoever, or remove this section of the manuscript.

The fundamental goal in this analysis was to use the information from the model parameters to predict lesion location in the brain. The underlying idea is that subjects with similar model parameters would also have similar lesions in their brains. The reviewer correctly points out that a formal cross-validation should ensure complete independence between subjects whose data go into constructing the predictive model, and the held-out subject whose lesion location is to be predicted.

We have addressed this concern in two ways. First, we removed the term “cross-validation” and now clarify that there is non-independence in the sources of the data from which the model is estimated, and the data that are predicted. We are thus using the term “prediction” in the more common way in which any correlation or regression model “predicts” one variable from another.

Second, we clarify that while there is non-independence in the source of data (that is, they come from the same subject), there **is** independence of the type of data; our prediction is thus not circular. Specifically, we are trying to predict a specific lesion pattern in the brain of one subject from other specific (and weighted) lesion patterns in the brains of other subjects. The predicted lesion pattern does not contribute in any way to the prediction map that we use to make these predictions; only the cognitive distance of the to-be-predicted patient is used to weight the other specific lesion patterns. This link between the cognitive profiles of one subject to all others is necessary as it gives meaning to the predictive relationship.

We have presented a more carefully qualified version of the analysis in our paper, because it is informative and could be of clinical utility. We note that complete independence (test data and model data do not come from the same subject) is impossible here, since this would amount to calculating the distance between model parameters just among the remaining subjects (without any reference to the left-out subject in question). Given our fundamental hypothesis about the similarity of model parameters and lesion location, this distance measure would then be essentially meaningless, because the similarity in model parameters is not tied to the left-out subject.

2. I was also asked by Dr. Horder to respond to a comment made by Reviewer 2 regarding another potential for bias in the predictive analysis. That reviewer pointed out that a hierarchical estimation procedure could create a bias in the cross-validation procedure because all parameter estimates were regularized with respect to a group mean. The authors responded in the last revision by changing their analysis to fit the model to each subject individually. I believe that this change should sufficiently remove the source of bias referenced by Reviewer 2.

3. In supplementary figures 3 and 4 demonstrating parameter recovery with this model, the figure legend describes the fitting as a hierarchical procedure. However, the authors are now using individual fits for the majority of their analyses since the last revision. At a quick glance, these figures look pretty much identical to those from the original submission where a hierarchical procedure was used. Given that removing the hierarchical procedure could also affect the fidelity of parameter recovery, it is important to complete this analysis with individual fits.

We recomputed this simulation without the hierarchical group distribution, which also reduces this analysis to a single simulation, because the different versions of modeling the group variance are now obsolete. We simulated 60 virtual subjects for each parameter combination, fitted each of them individually using Bayesian estimation, and averaged the maximum *a posteriori* point-estimates of the parameter distributions for the new Figure S3. Overall, the model is able to recover the true parameter values. However, just as in our hierarchical simulation, it tends to underestimate reward and punishment sensitivity in the medium and high parameter range, especially when the true decision consistency is low (indicating that choices are not strongly driven by the attention weights).

Minor:

1. P. 22, 'prediction analysis' 1st paragraph, the authors write that 'high distance values indicate that the profiles of model parameters between two participants are very similar.' However, for correlation distances, the opposite is true.

We use the original correlation between two parameter profiles as a distance measure in this study, not $(1 - \text{correlation})$ as it is often used. Therefore, a high value on this distance measures indicates high similarity in the parameter profiles.

2. P. 22, 'prediction analysis' 2nd paragraph, should say 'create' instead of 'created.'

This typo has been corrected.

3. Orange line for healthy subject performance in Figure 2 is missing for the F parameter.

These data were not provided in the original Bishara paper because in their model fitting the F parameter was fixed and set to 1. A brief explanation has been added to the legend of Figure 2.

4. Typo in legend for Figure S7 — 'VLSM' instead of 'VSLM.'

This typo has been corrected.

5. Panel labels are missing from Figure 7.

Panel labels were added to the figure.

Reviewers' comments:

Reviewer #4 (Remarks to the Author):

In this latest revision, Gläscher, Adolphs and Tranel have mostly made revisions to the description of their analysis testing the relationship of lesion location and model parameter similarity. In the previous version, the authors described an analysis where they used information about the distances of subjects in parameter space to make a predictive map for the location of a single subject's lesion. While described as a cross-validated procedure, this was not the case. I suggested that the authors either change this analysis so that it was genuinely cross-validated, or remove it from the manuscript. In this revision, the authors have opted to keep this section, while significantly walking back their claims about the outside validity of these results. They now describe this as a 'prediction analysis,' which amounts to an illustration of the multivariate relationship of model parameters and distributed brain damage. This prediction analysis is hard to judge on its own merits as it has been constructed in-house by the authors and has not been validated outside of this study.

The authors also do not indicate whether the observed lesion patterns in their two-step clustering solution were also observed in the cross-validation of this solution in the supplementary material, casting further doubt on the reliability of the behavior-lesion pattern mapping.

While the manuscript now appears to be far more accurate and balanced, the authors' original claims are substantially weakened. These factors have dampened my enthusiasm for the manuscript.

On another note: The authors say on p. 15 that previous studies of the effects of lesions on the WCST have not used VLSM, however, the authors themselves have reported VLSM results for this test in a previous PNAS paper (Gläscher et al., 2012), which does not appear to have been cited here. Relatedly, is there overlap in the data analyzed in that paper and the current study? There should be some acknowledgement that this manuscript is a re-analysis of previously published data if this is the case.

Minor:

Typo in Figure 2 caption: 'and outlier' should be 'an outlier.'

RE: NCOMMS-17-21473C

We thank the reviewer for the remaining comments, which we address here.

“In this latest revision, Gläscher, Adolphs and Tranel have mostly made revisions to the description of their analysis testing the relationship of lesion location and model parameter similarity. In the previous version, the authors described an analysis where they used information about the distances of subjects in parameter space to make a predictive map for the location of a single subject’s lesion. While described as a cross-validated procedure, this was not the case. I suggested that the authors either change this analysis so that it was genuinely cross-validated, or remove it from the manuscript. In this revision, the authors have opted to keep this section, while significantly walking back their claims about the outside validity of these results. They now describe this as a ‘prediction analysis,’ which amounts to an illustration of the multivariate relationship of model parameters and distributed brain damage. This prediction analysis is hard to judge on its own merits as it has been constructed in-house by the authors and has not been validated outside of this study.”

Response: We agree that our analysis is novel and innovative, in the sense that this particular multivariate analysis, and certainly on this particular data set, has not been done before. We disagree, however, with the claim that it “has not been validated outside of this study,” since the overall approach has certainly been validated outside this study. In fact, our analysis is basically an example of standard neuropsychological inference applied in many situations. In a nutshell: a clinician has available a multivariate pattern of task performances from neuropsychological assessment. One question that these data are used to answer is: what is the cause of the profile of task scores? A common inference is to diagnose a psychiatric and/or neurological cause, including dysfunction in particular brain regions. That is exactly the approach we take here. We have multivariate task data on many patients, we compare a single patient’s data to that, and we infer the patient’s lesion location based on the known associations between task data and the lesion locations of the other subjects.

Whether or not this approach works depends on many factors; but in our paper we show that it does work in our data. In the absence of a specific critique showing what the logical flaw in our approach is, we cannot agree with the reviewer’s critique here. Neither the fact that the analysis was done in-house, nor that this particular analysis is novel, strike us as a strong basis to doubt its validity. Again, we agree that if our logic were entirely novel, some additional validation might be warranted. But that is not the case here; this is a very straightforward prediction of the sort made all the time, just applied to novel data.

“The authors also do not indicate whether the observed lesion patterns in their two-step clustering solution were also observed in the cross-validation of this solution in the supplementary material, casting further doubt on the reliability of the behavior-lesion pattern mapping.”

Response: Although we have provided evidence for the reliability of our cluster solution using cross-validation in the previous version of the manuscript (see Figure S10), we agree with the reviewer that an estimate of the reliability of the *brain correlates* of our cluster solution will strengthen the original finding. To this end, we have recomputed the cross-validation analysis, updated Figure S10 (which gave almost identical results), and computed the mean percentage of overlap of the cross-validation density maps with the original density maps for each cluster. These data are now presented in Table S3 (reproduced below), which also lists the average number of subjects from the cross-validation samples that we included in a prior revision.

In general, the overlap between the cross-validation density maps and the original cluster density maps was moderate to high, especially for those clusters that were characterized by an impaired performance (clusters 1 and 2, and within 2, the sub-clusters 2a and 2c). The notable exception was the small overlap between the high-performance cluster number 3, probably because this cluster also contained the fewest number of patients.

Cluster No.	Full sample (divided by 2)	Cross-validation samples	Percent overlap of cross-validation density map
1	36 (18) ¹	19.3	73.3 %
2	266 (133)	131.4	87.3 %
2a	95 (47.5)	37.4	65.8 %
2b	93 (46.5)	52.6	47.4 %
2c	78 (39)	41.4	57.9 %
3	26 (13)	13.5	6.2 %

¹ The number in parentheses is half the number of subjects in the cluster to make it comparable with the cross-validation samples, which are 50% of the full sample size.

“While the manuscript now appears to be far more accurate and balanced, the authors’ original claims are substantially weakened. These factors have dampened my enthusiasm for the manuscript.”

Response: We feel that the numerous data we have added during the course of peer review, in both the manuscript proper and supplemental materials, have greatly strengthened the manuscript and have bolstered our arguments and conclusions.

“On another note: The authors say on p. 15 that previous studies of the effects of lesions on the WCST have not used VLSM, however, the authors themselves have reported VLSM results for this test in a previous PNAS paper (Gläscher et al., 2012), which does not appear to have been cited here. Relatedly, is there overlap in the data

analyzed in that paper and the current study? There should be some acknowledgement that this manuscript is a re-analysis of previously published data if this is the case.”

Response: The reviewer is correct that we have previously examined the WCST using VLSM (as well as several other tasks), and we now clarify this in our revised manuscript. There is some overlap in the raw data used in the two papers, and we acknowledge this also in the latest revision.

“Minor:

Typo in Figure 2 caption: ‘and outlier’ should be ‘an outlier.’

The typo has been corrected.

Reviewers' comments:

Reviewer #4 (Remarks to the Author):

In their response to my comments in the last round of revisions, the authors defended their prediction analysis as essentially being akin to the job of a neuropsychologist relating multivariate patterns of behavioral test scores to a neurological condition. However, this analogy doesn't seem to be accurate. In this manuscript, the authors appear to be making predictions about cluster membership, where clusters are defined by behavioral test scores (or, more accurately, model parameters). This analysis thus seems to be circular, as correlations between the parameters of the test case, and the parameters patients in a cluster (also defined by behavioral scores), are used to predict whether a patient belongs to this cluster. Instead, if the authors wanted to do the job of the neuropsychologist, they should be calculating the likelihood that a patient has damage at any given voxel (or cluster) given their multivariate pattern of model parameters, and using these likelihoods to predict the lesion location.

The authors also conducted a new analysis reporting the percentage of overlap of clusters derived from a randomly selected subset of the data with a clustering solution based on the whole dataset. They describe this analysis as a cross-validation procedure, but this characterization is not accurate. Instead of testing the validity of this clustering solution for an entirely separate set of subjects, the authors have tested whether this solution is apparent in a randomly selected half of the same dataset. As these data were included in the original clustering solution, these solutions are obviously not independent, and a moderately high overlap is not terribly surprising. Once again, there appears to be a problem with the authors' understanding of what constitutes a cross-validation analysis here.

I agree with the authors that the manuscript has been improved by the multiple rounds of revisions, in that many mistakes have been found and corrected (e.g. missing information about demographics and neuropsychological screening, the description of the prediction analysis as cross-validated, a failure to mention that this data has been analyzed previously). However, I cannot agree that their arguments have been bolstered in the process. What was originally presented as a completely original analysis of a large dataset with cross-validation of the main findings is instead something closer to a revisitation of older data, with much weaker claims about the robustness of these findings. I am also still unconvinced of the validity of the prediction analysis in testing what the authors claim and am bothered that the authors have once again mischaracterized an analysis as a 'cross-validation.' This kind of repeated carelessness has soured my view on the manuscript and makes me skeptical of whether it should be published in a high profile journal.

Lastly, Figures 7 and 2 did not appear in my copy of this version of the manuscript file for some reason. I assume these figures had not changed since the last revision and looked to that version while reviewing the manuscript.

Response to Reviewers:

[The authors indicated, to the editors, the removal of the prediction analysis from the manuscript, as earlier suggested by the reviewer. This included the removal of Figure 7 and Figure S12]